# A coarse-grained bacterial cell model for resource-aware analysis and design of synthetic gene circuits

Kirill Sechkar [1], Harrison Steel [1], Giansimone Perrino [2,3] ✉ & Guy-Bart Stan [2,3] ✉

Within a cell, synthetic and native genes compete for expression machinery, influencing cellular process dynamics through resource couplings. Models that simplify competitive resource binding kinetics can guide the design of strategies for countering these couplings. However, in bacteria resource availability and cell growth rate are interlinked, which complicates resource-aware biocircuit design. Capturing this interdependence requires coarse-grained bacterial cell models that balance accurate representation of metabolic regulation against simplicity and interpretability. We propose a coarse-grained *E. coli* cell model that combines the ease of simplified resource coupling analysis with appreciation of bacterial growth regulation mechanisms and the processes relevant for biocircuit design. Reliably capturing known growth phenomena, it provides a unifying explanation to disparate empirical relations between growth and synthetic gene expression. Considering a biomolecular controller that makes cell-wide ribosome availability robust to perturbations, we showcase our model's usefulness in numerically prototyping biocircuits and deriving analytical relations for design guidance.

The engineering of novel biological systems with useful applications, known as synthetic biology, promises to address global challenges by revolutionizing healthcare, agriculture, and manufacturing[1]. While any engineered system's behavior must be predictable, its forecasting for synthetic biology designs is complicated by the interconnectedness of biological processes in living cells[2–4]. Indeed, even the gene circuit components that do not engage in direct biological interactions may exhibit interdependence caused by indirect couplings via shared cellular resources. Specifically, resource competition couplings arise when heterologous genes introduced into a host cell compete among themselves and with native genes for the same finite pools of resources that enable gene expression. Due to this competition, a synthetic gene circuit's performance may not be reliably predicted based on the characterization of its constituent modules in isolation[5,6]. Moreover, the redirection of resources from native gene expression, known as

gene expression burden, hinders cell growth and biomass accumulation[3,4]. Resource couplings therefore present a key challenge in the engineering of bacterial, fungal, and mammalian cells[7–9].

To mitigate resource couplings, synthetic gene expression can be kept low enough to have no significant effect on cellular homeostasis[10]. Alternatively, feedback loops can improve a circuit's robustness to perturbations such as resource couplings[7]. Moreover, synthetic genes can be expressed via orthogonal molecular machinery which is unused in native gene expression and forms a separate resource pool, thereby reducing crosstalk between engineered circuits and native processes[11]. Nonetheless, the low-expression strategy is unsuitable for applications requiring high protein production, while the synthesis of orthogonal machinery or regulator proteins enabling feedback control can itself burden the cells[10]. Mathematical modeling of resource competition represents a promising approach to resource-aware design of

[1]Department of Engineering Science, University of Oxford, Parks Road, Oxford OX1 3PJ, UK. [2]Department of Bioengineering, Imperial College London, South Kensington Campus, London SW7 2AZ, UK. [3]Imperial College Centre of Excellence in Synthetic Biology, Imperial College London, South Kensington Campus, London SW7 2AZ, UK. ✉e-mail: g.perrino@imperial.ac.uk; g.stan@imperial.ac.uk

biocircuits and the development of more sophisticated strategies for countering resource couplings.

One can explicitly model all stages of a substrate binding and unbinding the shared resource for which it competes with other substrate species. However, gene expression models can also be simplified to ensure better interpretability and lower computational complexity. The entire resource-dependent process can be described with an effective rate constant that reflects the concentrations of all competing substrates (Fig. 1a), assuming that very fast association and dissociation make a resource-substrate complex's concentration change very little on the time scale of other reactions in the cell[9,12,13]. Such simplified models allow to easily determine a genetic module's sensitivity to resource couplings[6] and to optimize design parameters like gene dosage and ribosome-binding site (RBS) strength, achieving desired outputs despite unwanted couplings[12,14]. Modeling insights can also help design circuits that efficiently mitigate unwanted couplings[7,13] or even leverage resource competition as a gene regulation mechanism to achieve other objectives, such as plasmid copy number-independent synthetic gene expression in mammalian cells[15].

The analysis of resource couplings in bacteria is further complicated by the interplay between synthetic gene expression and cell growth rate. Bacterial cells are fast-growing, so dilution due to cell division significantly influences gene expression dynamics[16]. Accordingly, changes in the cell's growth rate caused by gene expression burden can qualitatively alter a gene circuit's behavior[16-18]. Moreover, the size of a bacterial cell's pool of gene expression machinery—specifically, ribosomes—is also related to its growth rate[19,20]. This interdependence, described by the experimentally observed "bacterial growth laws"[21] (Fig. 1b), arises because cells optimize their gene expression to maximize the steady-state growth rate in given environmental conditions[22-25]. These phenomena limit the predictive power of simple resource competition models, which assume constant growth rate and resource availability[6,9,10]. The development of bacterial resource-aware biocircuits thus calls for resource-aware models that consider synthetic circuits within the context of the host cell and account for the impact of burden on cell growth.

Since allocating resources between different genes to achieve maximum growth rate can be considered an optimization problem, solving it allows to approximately predict gene expression in certain conditions[23,26,27]. However, in reality living cells do not behave as ideal growth rate optimizers, but rather implement near-optimal gene regulation strategies via biological reactions[23]. Although some past studies have reproduced the bacterial growth laws by assuming constitutive ribosome expression[28,29], evidence increasingly suggests that near-optimal bacterial resource allocation is powered by the regulation of ribosomal genes' transcription by guanosine tetraphosphate (ppGpp), an alarmone molecule whose concentration lets the cell perceive its growth rate[30,31].

Understanding how biochemical signals reflect cell growth rate and enable the optimization of gene expression requires mechanistic cell models, which incorporate the trade-offs faced by living cells, such as the finiteness of the cell's mass and its pool of gene expression machinery, energy, protein synthesis precursors, and other resources[28,32]. A range of such models with different levels of granularity exists, from whole-cell models[33] considering all known cellular processes to low-dimensional models, which consider the cell as a simple self-replicating machine that produces protein biomass in a single-step reaction. In the latter case, the cell's proteins with similar function and expression dynamics are grouped into several coarse-grained classes, and resource allocation modeling amounts to considering the ratios of different protein classes' mass fractions in the overall biomass[22,23,26,34-37]. Modeling frameworks that lie between these two extremes explicitly consider some aspects of gene expression and metabolic regulation—typically those believed most relevant for the phenomena they aim to explain—and adopt a simplified view of the others[27-29,38-40].

The investigation of a host cell's interactions with synthetic gene circuits requires a balance between model realism and minimum complexity. Indeed, abstracting some gene expression steps or cellular signaling pathways carries the risk of neglecting important ways in which a synthetic device can influence the cell. Conversely, a finer view of gene expression and metabolic regulation, as well as the incorporation of biochemical interactions that are only significant in certain culture environments, may yield an overly detailed model with many unknown or unidentifiable parameters[41]. Excessive model complexity can also hinder informative biocircuit analysis and complicate the understanding of core biochemical processes defining the cell's state[28].

In this study, we aimed to enable easy yet reliable resource- and cellular context-aware design of synthetic circuits by developing a coarse-grained mechanistic model that would:

(1) Based on physiologically meaningful parameters, allow to model the expression of a biocircuit's genes, considering their interactions via both indirect resource couplings and direct mechanisms commonly employed in synthetic biology.

(2) Account for the context of the host cell and its interaction with synthetic genes, incorporating the key regulatory pathways

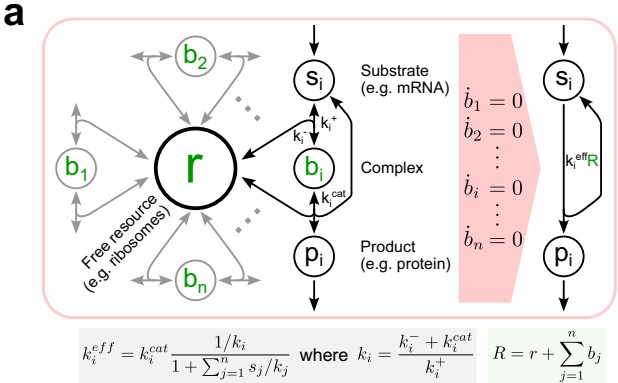
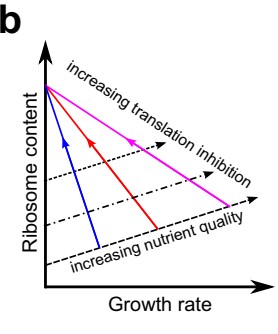

$$k_i^{eff} = k_i^{cat} \frac{1/k_i}{1 + \sum_{j=1}^{n} s_j/k_j} \quad \text{where} \quad k_i = \frac{k_i^- + k_i^{cat}}{k_i^+} \qquad R = r + \sum_{j=1}^{n} b_j$$

**Fig. 1 | Models and empirical laws for cellular resource availability. a** Instead of explicitly modeling the binding and unbinding of all competing substrates to the free resource (left), one can define the rate of a compound's synthesis as a product of the total abundance of the resource $R$, which can be free ($r$) or bound to a substrate molecule ($b_i$), and the effective rate constant $k_i^{eff}$ (right). The effective rate constant depends on all competing species' concentrations, affinities to the resource, and product synthesis rates[9]. **b** First (dashed line) and second (solid lines) bacterial growth laws relate growth rate to ribosome content in different conditions. Formulated by Scott et al.[21], they postulate that the cell's ribosome content increases linearly with the growth rate as the culture medium's nutrient quality improves, but this relationship becomes inverse when translation is inhibited by an antibiotic.

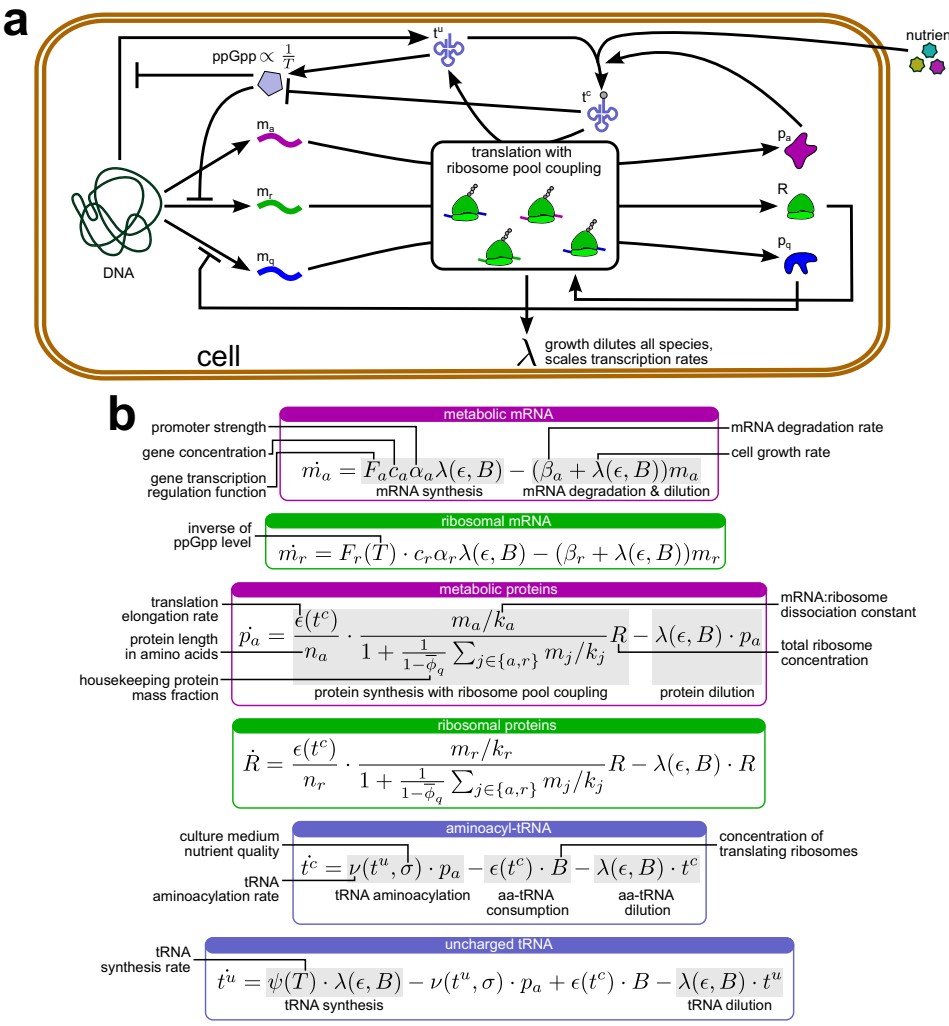

**Fig. 2 | Schematic and definition of the proposed coarse-grained resource-aware cell model. a** Schematic of our host cell model. mRNAs are transcribed from genomic DNA and translated to make proteins. Competitive ribosome binding is not modeled explicitly—instead, effective rate constants relate each gene's translation rate to the concentrations of protein precursors, ribosomes and all mRNAs in the cell. Ribosomes enable translation, whereas metabolic proteins catalyze nutrient import and tRNA aminoacylation. Native homeostatic regulation maintains housekeeping protein expression levels constant. The synthesis of tRNAs and ribosomal mRNAs is repressed by ppGpp, whose concentration reflects the reciprocal of the ratio of concentrations of charged and uncharged tRNA concentrations. The overall rate of protein synthesis defines the growth rate, since dilution due to growth must keep the cell's total protein mass constant. **b** The model's ordinary differential equations (ODEs) and the physiological meaning of their terms. To avoid clutter, parameters and terms for different genes but with the same meaning are annotated only once.

presently understood to underlie cellular resource allocation and growth control, particularly ppGpp signaling.

(3) Be minimally complex, enabling the derivation of informative analytical relations which can guide the choice of a biocircuit's design parameters to achieve a desired behavior.

Defining such a model and parameterizing it for *E. coli*, we show that it reproduces experimentally observed bacterial growth phenomena, as well as empirical relations between the burden-dependent reduction in growth rate and different quantities characterizing heterologous gene expression in the cell. We also use our model to numerically reproduce the experimentally observed effects of resource competition on the behavior of self-activating gene circuits. Finally, we showcase our model's usefulness for the development of resource-aware biocircuits by leveraging it to propose and analyze a biomolecular controller for mitigating gene expression burden. By maintaining near-constant ribosome availability at a cell-wide level, it reduces the effect of indirect couplings via the shared ribosome pool.

## Results

### A resource-aware cell model predicts growth phenomena

Our model, depicted in Fig. 2, distinguishes three bacterial gene classes, respectively labeled $r$, $a$, and $q$: ribosomal, metabolic, and housekeeping. The metabolic gene class enables tRNA aminoacylation and includes both aminoacyl-tRNA synthetases themselves and the enzymes enabling import and conversion of the culture medium's nutrients into protein precursors. Other non-ribosomal genes belong to the housekeeping class, whose expression is regulated to be independent of culturing conditions[19,20,26]. Following a coarse-grained approach, all metabolic genes are treated as a single lumped gene; the same strategy is applied to ribosomal genes. Meanwhile, the abundance of housekeeping proteins in the cell is assumed constant—under a wide range of conditions, their share in the cell's protein mass is fixed at $\overline{\phi}_q \approx 0.59$[20]. Hence, we avoid modeling their expression explicitly (Supplementary Note S1.5). Besides mRNA and protein expression, we model the concentrations of uncharged and charged (i.e., aminoacylated) tRNA molecules in the cell, since they play a key role in determining translation rates

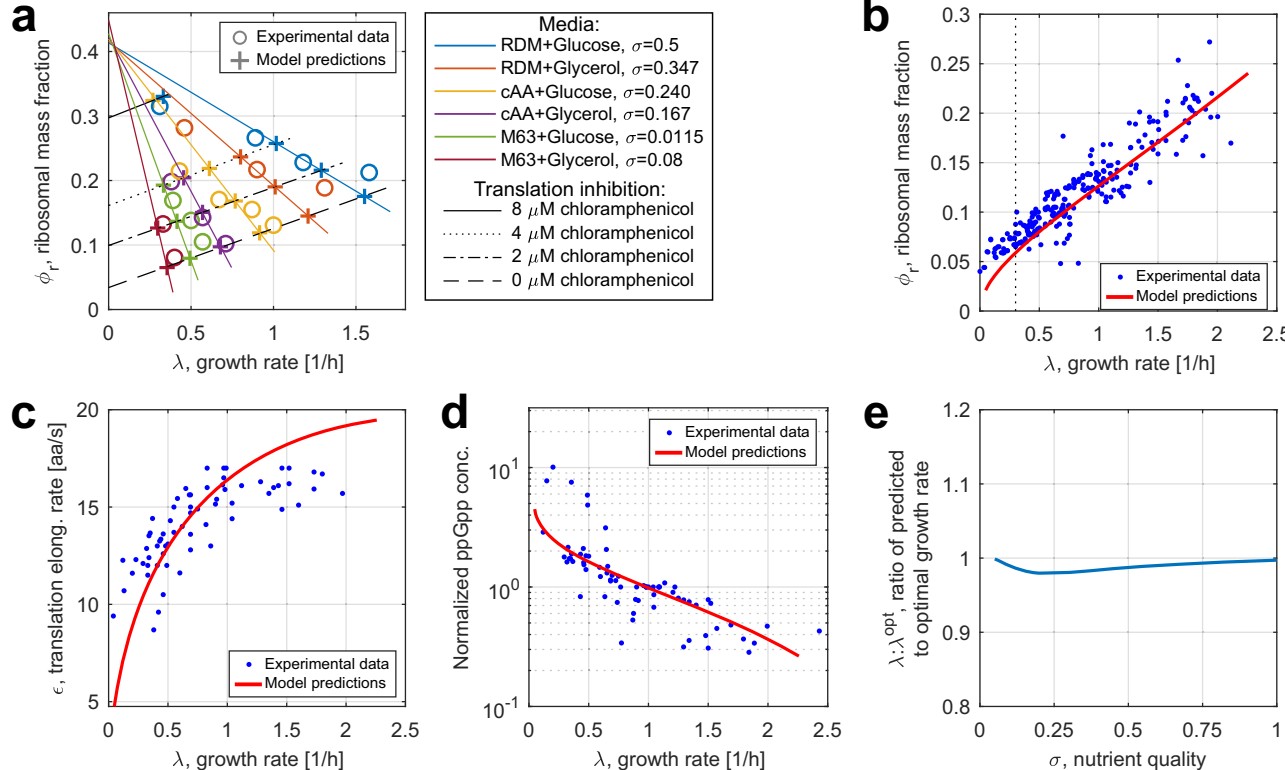

**Fig. 3 | Growth phenomena predicted by our model. a** Experimental measurements by Scott et al.[19] used in parameter fitting (circles) and model predictions (crosses) for steady-state ribosomal mass fractions and growth rates. The first and the second growth laws are illustrated by varying culture media and extents of translation inhibition, respectively. Second growth law fits do not include the two predicted points for translation inhibition with 8 µM of chloramphenicol due to them diverging from the linear growth law trend. Chloramphenicol action was modeled as outlined in Supplementary Note S2.1. **b–d** Comparison of model predictions (obtained by simulating Eqs. (15)–(20)) for the cell in steady state with experimental data from previous studies[26]. In (**b**), the dotted vertical line denotes the $\overline{\lambda} = 0.3$ threshold, left of which model predictions for $\overline{\phi}_r$ significantly diverge from the experimental data. In (**d**), the ppGpp levels are normalized to the reference value for which $\overline{\lambda} \approx 1\,\text{h}^{-1}$. **e** Ratio of the steady-state growth rate $\overline{\lambda}$ predicted by our model to the optimal growth rate $\overline{\lambda}^{opt}$ for $\sigma$ varied from 0.05 to 1. For each medium nutrient quality $\sigma$ considered, we did the following. First, we recorded the steady-state cell growth rate $\overline{\lambda}$ (here, the bar notation indicates a variable's steady-state value) predicted by our model. According to it, $T$, the inverse of ppGpp concentration, reflects the charged and uncharged tRNA levels as detailed by Eq. (28). Then, we assumed that ppGpp concentration in the cell is instead constant, ran simulations for different fixed values of $T$ and identified the maximum steady-state cell growth rate $\overline{\lambda}^{opt}$ across all considered values. Source data are provided as a Source Data file.

and resource allocation between gene classes[31]. Our modeling choices and derivations are detailed in "Methods" and Supplementary Note S1.

To validate model predictions against experimental data, we parameterized it for *E. coli*, the most studied bacterial model organism[19,26,31,42] and one of the most popular host organisms in synthetic biology[43]. Most parameter values, displayed in Supplementary Table 1, were taken from literature. The rest were fitted to experimental measurements of growth rate and ribosomal mass fraction of *E. coli* subjected to different concentrations of the translation-inhibiting antibiotic chloramphenicol in various growth media[19,26] as described in "Methods" and Supplementary Note S2.

As demonstrated by Fig. 3a, our model's steady-state predictions for different culture conditions are generally consistent with empirical bacterial growth laws illustrated in Fig. 1b[19,21]. Indeed, when nutrient quality improves and chloramphenicol levels remain unchanged, the cell's ribosome content increases linearly with the growth rate, obeying the first bacterial growth law. Moreover, augmenting translation elongation inhibition for the same nutrient quality produces an inverse proportionality between the ribosome content and growth rate for chloramphenicol levels of up to 4 µM. We also confirmed the consistency of our model predictions for the cell's ribosome content, translation elongation rate, and ppGpp level with experimental results. To this end, we varied the nutrient quality factor $\sigma$ without inhibiting translation and compared our model's steady-state predictions with

experimental data from the 37 different experimental studies compiled by Chure and Cremer[26] (Fig. 3b–d). As for the cell's dynamic behavior, in Supplementary Note S1.6 we compared experimental data with the predictions of our model's extension that allows to simulate nutrient upshift scenarios[34,36].

Figure 3b supports the linear-like bacterial growth law dependence between ribosome content and growth rate, although for growth rates below $\lambda \approx 0.3\,\text{h}^{-1}$ our model predictions take a nonlinear downturn, diverging from experimental measurements. Moreover, for *E. coli* subjected to 8 µM of chloramphenicol, predictions do not strictly follow the linear trends. Notably, these discordances arise in highly unfavorable conditions—that is, strong inhibition of translation or very poor nutrient quality of the medium. Hence, to some extent they may be attributed to experimental errors, since it is difficult to ensure that measurements in such conditions pertain to cells in steady state[26]. Hostile environments can also trigger the cell's stress response mechanisms unconsidered by our model, such as ribosome inactivation, preserving the bacterium's ability to synthesize proteins in adverse conditions[42]. Furthermore, since mRNA synthesis rate depends on the cell's growth rate[44], low-nutrient culture media reduce the overall concentration of transcripts in the cell. This leads to mRNA scarcity, rather than translational resource allocation that our model focuses on, becoming the key determinant of the growth rate[27]. Together, these factors limit our model's predictive power in unfavorable conditions.

Finally, in Fig. 3e we confirmed whether ppGpp regulation in our model reproduces near-optimal control of steady-state growth rates in a wide range of culturing conditions in line with prior modeling and experimental studies[23,24,26]. As shown in Fig. 3e, for $0.05 \leq \sigma \leq 1$ the growth rate forecast by our ppGpp regulation model was always very close (within 2.3%) to the optimal value, which supports the notion of near-optimal control of resource allocation.

## Including heterologous gene expression in the cell model

Our cell model can be extended to consider heterologous gene expression and simulate gene circuit dynamics while accounting for resource couplings and the effects of heterologous gene expression on the host's metabolism that can influence the biocircuit's performance, such as growth rate changes. Besides native genes, let there be a set of $L$ heterologous genes $X = \{x_1, \ldots, x_L\}$. To model their expression, we add to Eqs. (15)–(20) a pair of Ordinary Differential Equations (ODEs) for each heterologous gene, which describes its transcription and translation (Supplementary Note S3.1). For a synthetic gene $x_l$, they are:

$$\dot{m}_{x_l} = F_{x_l} c_{x_l} \alpha_{x_l} \lambda(\epsilon, B) - (\beta_{x_l} + \lambda(\epsilon, B)) m_{x_l} \tag{1}$$

$$\dot{p}_{x_l} = \frac{\epsilon(t^c)}{n_{x_l}} \cdot \frac{m_{x_l}/k_{x_l}}{1 + \frac{1}{1-\overline{\phi}_q} \sum_{j \in \{a,r\} \cup X} m_j/k_j} R - \lambda(\epsilon, B) \cdot p_{x_l} \tag{2}$$

where all parameters have a similar definition to those of the parameters describing the native genes, while the transcription regulation function $F_{x_l}$ is circuit- and gene-specific. Likewise to the original host cell model, we assume that transcriptional resource couplings are insignificant, as are the toxicity and active degradation of heterologous proteins, although the latter assumption can be lifted by making minor modifications as outlined in Supplementary Note S3.2. Meanwhile, the constant housekeeping protein mass fraction $\overline{\phi}_q \approx 0.59$ remains the same despite synthetic gene expression[19–21]. Consequently, the model's original ODEs for native mRNA and tRNA levels are not altered by the addition of synthetic genes. Conversely, Eqs. (17) and (18) describe protein concentrations and thus include terms for translation. Therefore, they must be amended to consider additional synthetic mRNAs competing for ribosomes:

$$\dot{p}_a = \frac{\epsilon(t^c)}{n_a} \cdot \frac{m_a/k_a}{1 + \frac{1}{1-\overline{\phi}_q} \sum_{j \in \{a,r\} \cup X} m_j/k_j} R - \lambda(\epsilon, B) \cdot p_a \tag{3}$$

$$\dot{R} = \frac{\epsilon(t^c)}{n_r} \cdot \frac{m_r/k_r}{1 + \frac{1}{1-\overline{\phi}_q} \sum_{j \in \{a,r\} \cup X} m_j/k_j} R - \lambda(\epsilon, B) \cdot R \tag{4}$$

Combined, all extensions yield Model VI in Supplementary Note S3.1, which describes the cell and the synthetic circuitry it hosts. All synthetic genes' expression in this work was modeled using these ODEs, with the generic Supplementary Eqs. (75) and (76) for synthetic gene expression substituted by the specific ODEs for each circuit, which we explicitly provide in Supplementary Note S4.

To demonstrate how our modeling framework captures the implications of resource competition for synthetic gene expression, we recreated two experimentally documented cases of self-activating gene circuits' behavior being qualitatively altered by resource competition and the dependence of host cell growth on synthetic gene expression. First, we considered the "winner-takes-all" phenomenon[5], the ODEs for which are provided in Supplementary Note S4.2.1. Alone in the host cell, a self-activating synthetic gene can act as a bistable switch with a high- and a low-expression stable steady states[45]. In the winner-takes-all scenario, two such switches in the same cell interact via the shared resource pool (Fig. 4a). Hence, if one switch

("the winner") reaches its high-expression equilibrium first, increased resource competition can prevent the other switch from ever reaching the corresponding high-expression steady state (Fig. 4b and Supplementary Fig. 4b–d).

Second, we reproduced the emergence of bistability due to host-circuit interactions[46,47] using the equations in Supplementary Note S4.3.1. Typically, self-activating genes act as bistable switches only if positive feedback is cooperative, i.e., a protein activates gene expression more efficiently if multiple copies of it are present. A heterologous T7 RNA polymerase (RNAP) transcribing its own gene exhibits non-cooperative self-activation (Fig. 4c), so its expression is expected to be monostable. However, by slowing the host cell's growth rate and thus protein and mRNA dilution, burden introduces an additional feedback loop which can confer bistability to this circuit despite its non-cooperativity as shown in Fig. 4d, e.

Nonetheless, simulations and bifurcation analysis[48] reveal that the winner-takes-all effect fades as the self-activating genes' resource demand is reduced (Supplementary Fig. 4e), whereas changing T7 RNAP's toxicity can render the non-cooperative self-activator monostable (Supplementary Fig. 5). Our model can therefore be used to determine whether given circuit design parameters can give rise to resource competition phenomena of interest.

## Analytical predictions reveal how burden affects the cell

Besides enabling numerical prediction of circuit behavior, with some simplifying assumptions our model allows to derive analytical relations capturing the effect of different parameters on different variables' steady-state values. Namely, we assume that heterologous gene expression burden has very little effect on the cell's steady-state translation elongation rate and ribosomal gene transcription regulation function. While no experimental studies to date have directly evaluated these values' burden-dependence, in Supplementary Note S3.4 we show that this assertion directly follows from our model's definition, as well as confirm it numerically. This allows to postulate, regardless of which synthetic genes are present, that $F_r \approx \overline{F}_r^{NB}$ and $\epsilon \approx \overline{\epsilon}^{NB}$, as well as $k_i \approx \overline{k}_i^{NB}$ because the mRNA-ribosome dissociation constants are defined as functions of $\overline{\epsilon}^{NB}$. The $^{NB}$ ("no burden") index denotes steady-state values in absence of any synthetic gene expression, i.e., $X = \varnothing$. While the formulae derived in this section require knowing the values of $\overline{\epsilon}^{NB}$, $\overline{F}_r^{NB}$, and $\overline{k}_i^{NB}$, they can easily be retrieved by simulating the host cell model without synthetic genes for a given $\sigma$. Additionally, we assume that all mRNA molecules in the system decay at roughly the same rate, i.e., $\beta_i \approx \beta_j, \forall i, j \in \{a, j\} \cup X$. Importantly, while the transcripts of individual native *E. coli* genes may have very different degradation rates[49], the coarse-grained nature of our model means that this assumption only concerns the average degradation rates across the native gene classes, each of them spanning many genes.

Resource coupling analysis is commonly facilitated by lumping a gene's parameters into a coefficient which quantifies the gene's ability to seize expression resources and its own susceptibility to competition from other genes[6,12,29]. For a synthetic gene $x_l \in X$, we define it as the "translational burden" $\xi_{x_l}$ shown in Eq. (5), where $\overline{F}_{x_l}$ is the steady-state value of gene $x_l$'s transcription regulation function:

$$\xi_{x_l} = \frac{\overline{F}_{x_l} c_{x_l} \alpha_{x_l}}{\overline{k}_{x_l}^{NB}} \tag{5}$$

As revealed by the derivations in Supplementary Note S3.5, our translational burden factor combines the advantages of several existing lumped resource competition quantifiers. By considering the host cell's context, similarly to the "resource recruitment strengths" $J$ defined by Santos-Navarro et al.[29], we can use $\xi_{x_l}$ values to capture the competition between synthetic and native genes. Namely, the steady-

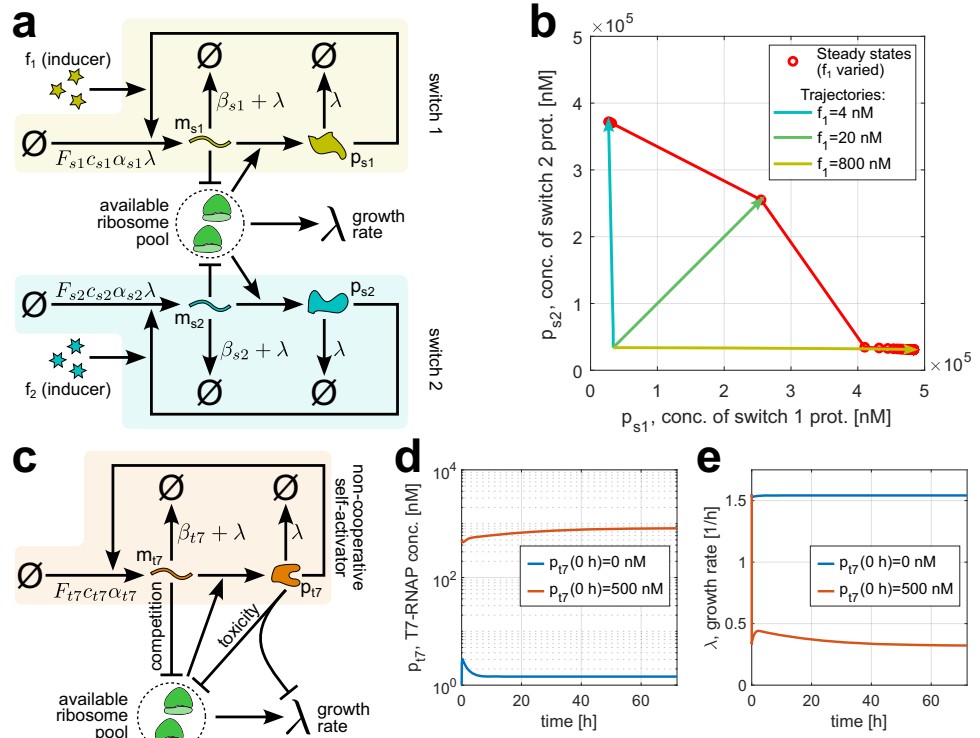

**Fig. 4 | The model captures how resource competition and host-circuit interactions qualitatively alter circuit behavior. a** In the winner-takes-all scenario, the cell hosts two bistable switches, each consisting of a protein that, upon being allosterically modulated by an inducer molecule, cooperatively acts as a transcription factor activating its own expression. Adding the corresponding inducer to the medium causes a switch to move toward a high-expression equilibrium. For higher inducer levels, this happens on a shorter timescale (Supplementary Fig. 4a)[5]. **b** Phase plane diagram showing the two-switch system's behavior upon simultaneous addition of inducer 1 and inducer 2 to the culture medium. Thin lines with arrowheads: the system trajectories for different concentrations of inducer 1 being added. Circles joined by a line: system's final steady states as the concentration of inducer 1 increases from 4 nM to 800 nM with a step of 4 nM. In all cases, the concentration of inducer 2 being added is 20 nM. In line with winner-takes-all

behavior, the switch with a lower inducer concentration (slower activation) is prevented from reaching a high-expression equilibrium. Co-activation is achieved when the timescales of activation are identical for both switches, i.e., $f_1 = f_2 = 20$ nM[5]. **c** A heterologous T7 RNAP transcribing its own gene exhibits non-cooperative self-activation. However, bistability arises due to the host cell growth rate's susceptibility to resource competition and synthetic protein toxicity. Note that here the gene transcription rate does not scale with cell growth due to transcription being enabled by heterologous machinery[46]. **d, e** Depending on the initial condition, RNA polymerase concentration and cell growth rate converge to different steady-state values, indicative of bistability. For both trajectories shown, the initial condition for heterologous mRNA levels is $m_{t7}(0\,h) = 0$. The parameters and ODEs used in simulations are given in Supplementary Notes S4.2.1 and S4.3.1 for (**b**) and (**d, e**), respectively. Source data are provided as a Source Data file.

state mass fraction of heterologous protein in the cell as a function of $\xi$ can be estimated as:

$$\overline{\phi}_X(\xi) \approx \frac{\xi}{\xi + \left(\sum_{j \in \{a,r\}} \overline{F}_j^{NB} c_j \alpha_j / \overline{k}_j^{NB}\right)} \quad \text{where} \quad \xi = \sum_{x_l \in X} \xi_{x_l} \quad (6)$$

However, unlike resource recruitment strengths and similarly to the "resource demand coefficients" $Q$ used by McBride and Del Vecchio[6], $\xi_{x_l}$ is independent of the cell growth rate. Hence, the growth rate itself can be analytically estimated from synthetic gene parameters according to Eq. (7):

$$\overline{\lambda}(\xi) \approx \frac{\overline{\epsilon}^{NB}(1 - \overline{\phi}_q)}{M} \cdot \frac{\overline{F}_r^{NB} c_r \alpha_r / \overline{k}_r^{NB}}{\xi + \left(\overline{F}_r^{NB} c_r \alpha_r / \overline{k}_r^{NB} + c_a \alpha_a / \overline{k}_a^{NB}\right)} \quad (7)$$

A Hill relationship akin to Eq. (7) is sometimes used in resource-aware models that abstract the host's native gene expression yet aim to capture cell growth rate's burden-dependence[17,18]. However, these relations involve scaling factors defined arbitrarily for each synthetic circuit, such as the "metabolic burden threshold" in ref. 18. Conversely, our expression only involves the physiological parameters of native and synthetic genes.

Furthermore, the change in the cell growth rate relative to $\overline{\lambda}^{NB}$ – that is, $\overline{\lambda}(\xi = 0)$ – can be related to the total mass fraction of all heterologous proteins in the cell $\overline{\phi}_X$ (Supplementary Note S3.6):

$$\frac{\overline{\lambda}}{\overline{\lambda}^{NB}} \approx 1 - \frac{\overline{\phi}_X}{1 - \overline{\phi}_q} \quad (8)$$

Several different empirical relations, such as Hill or linear dependencies, have been suggested to link gene expression burden to the reduction in cell growth rate[6,19]. Therefore, previous works have considered different formulae as part of their gene expression models in order to ensure that the modeling outcomes stay valid regardless of the assumed burden-growth dependency[50]. However, Eqs. (7) and (8) hint that these relations may not be mutually exclusive, but rather apply to different quantities describing heterologous gene expression. Namely, Eq. (7) relates growth rate to the translational burden—that is, a measure of the synthetic genes' resource demand, determined by the circuit's design parameters—giving rise to a Hill relation akin to that used in the work of McBride et al.[6]. If growth rate is linked to the heterologous protein yield, which is affected by resource couplings, a linear dependency given by Eq. (8) emerges, matching the observations of Scott et al.[19]. Our model therefore provides a possible unifying framework to explain different empirical laws describing the dependence of cell growth rates on synthetic gene expression.

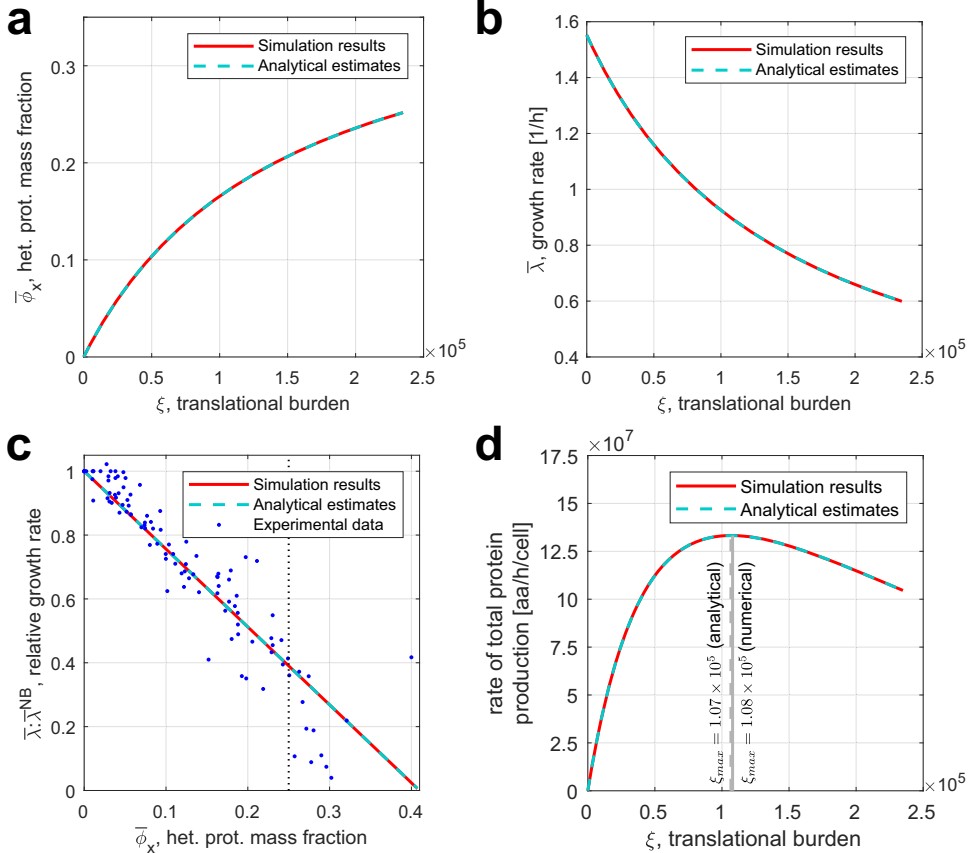

**Fig. 5 | Numerical and analytical predictions for heterologous gene expression and its effects on the host cell. a, b** Hill activation and repression functions respectively relate translational burden to the steady-state heterologous protein mass fraction $\overline{\phi}_X$ and cell growth rate $\overline{\lambda}$. **c** In the steady state, $\overline{\phi}_X$, the mass fraction of heterologous proteins in the cell, is linearly related to $\overline{\lambda} : \overline{\lambda}^{NB}$, the ratio of the corresponding growth rate to that without burden. The dotted vertical line denotes the $\overline{\phi}_X = 0.25$ threshold, right of which model predictions become unreliable. **d** The rate of total protein production per cell as a function of the translational burden. $\xi_{max}$ (vertical lines) is the burden value maximizing heterologous protein synthesis. All simulations assume $\sigma = 0.5$; the parameters and ODEs used in simulations are given in Supplementary Note S3.1. Source data are provided as a Source Data file.

To compare our analytical estimates with the model's numerical predictions for different extents of translational burden, we simulated the expression of a constitutive heterologous gene by an *E. coli* cell (parameterized in Supplementary Table 4). The burden $\xi$ exerted on the host by the synthetic gene was varied by sweeping through different values of the gene's DNA concentration from 1 nM to 1100 nM, producing the results plotted in Fig. 5a–c. Our approximate analytical predictions of Eqs. (6)–(8) closely follow the steady-state values obtained by numerical simulation (Fig. 5a–c, respectively), vindicating the assumptions made in the course of our analytical derivations. In Fig. 5c, both the analytical and numerical predictions are largely concordant with experimental data compiled by Chure and Cremer[26], although the real and predicted measurements diverge when the heterologous protein overexpression stress is very high ($\overline{\phi}_X > 0.25$). The reasons for this are likely similar to the general reasons why our model's predictions do not match experimental data in highly unfavorable conditions—namely, measurement errors[26], ribosome inactivation[42], and the dependence of slowly dividing cells' growth rate on their mRNA content[27].

Analytical relations derived using our model and simplifying assumptions can facilitate the design of heterologous gene expression systems. For example, consider a population of $N$ *E. coli* cells expressing a constitutive heterologous protein of interest $p_{poi}$ and dying at a constant rate $\delta$, modeled by Eq. (9):

$$\dot{N} = \lambda N - \delta N \qquad (9)$$

We can analytically find the optimal translational burden $\xi_{max}$ that maximizes the production rate $\mu$ of the protein of interest by the cell population (Supplementary Note S3.7). This yields an analytical optimality condition based on the gene of interest's design parameters, namely its DNA copy number $c_{poi}$, promoter strength $\alpha_{poi}$, and apparent mRNA-ribosome dissociation constant $\overline{k}_{poi}^{NB}$ (reflecting the RBS strength):

$$\frac{c_{poi}\alpha_{poi}}{\overline{k}_{poi}^{NB}} = \xi_{max} = \frac{1 - \delta/\overline{\lambda}^{NB}}{1 + \delta/\overline{\lambda}^{NB}} \cdot \sum_{j \in \{a,r\}} \frac{\overline{F}_j c_j \alpha_j}{\overline{k}_j^{NB}} \qquad (10)$$

In Fig. 5d, we plot the total protein production rate (calculated according to Supplementary Eq. (107) in Supplementary Note S3.7) as a function of the gene expression burden $\xi$, assuming that $\delta = 0.25\,\text{h}^{-1}$. Notably, $\xi_{max}$ yielded by Eq. (10) lies within 0.93% of the numerically found optimal value.

## Mitigating cell-wide resource couplings by integral feedback

To demonstrate how our model can facilitate resource-aware circuit development, we used it to design and analyze a biomolecular controller that mitigates resource couplings via the shared ribosome pool, reducing the impact of gene expression burden. Usually, robustness to couplings is achieved for a single variable or a limited set of synthetic genes that draw resources from an orthogonal pool, whose size is regulated by a biomolecular controller[7]. Conversely, our design reduces fluctuations in resource availability at the whole-cell level. Our cell

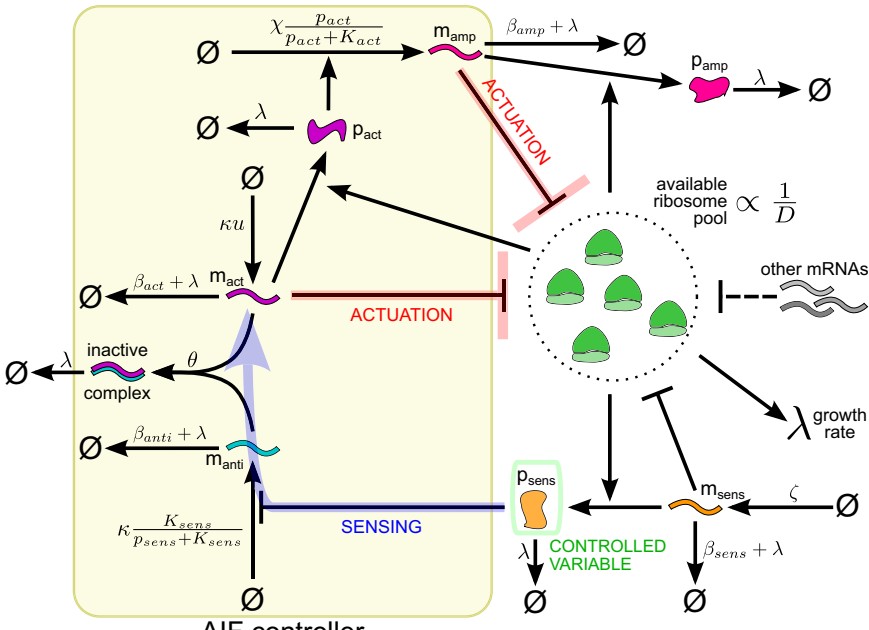

**Fig. 6 | Schematic of the integral biomolecular controller mitigating resource couplings.** If some disturbance increases the number of mRNAs competing for ribosomes, the concentration of the constitutive protein $p_{sens}$ falls, which activates the production of annihilator RNA $m_{anti}$. Hence, the concentration of the actuator mRNA $m_{act}$ decreases; so does the expression of the amplifier mRNA $m_{amp}$, regulated by the actuator protein $p_{act}$. The ribosomes previously sequestered by the amplifier and actuator mRNAs are therefore released, increasing the resource availability back to its original level.

model is particularly suited for studying such controllers: by considering the expression of not just synthetic but all genes in the cell, it captures cell-wide resource couplings that our controller seeks to minimize. The analysis of these couplings is facilitated by the cell model's effective rate constant framework.

The proposed circuit, illustrated in Fig. 6 and modeled by Supplementary Eqs. (121)–(127), employs the Antithetic Integral Feedback (AIF) motif, which can maintain a physiological variable of interest at a desired setpoint value[51,52]. It comprises an annihilator and an actuator species, where the annihilator's synthesis rate depends on the controlled variable and the actuator species is produced at a constant rate, which sets the desired reference value. Since one annihilator and one actuator molecule can react to disable each other, the concentration of remaining non-annihilated actuator molecules reflects the integral of the error between the controlled variable and the reference. By influencing the rest of the system, the actuator molecules ensure robust perfect adaptation (RPA) to disturbances[51,52]. To control ribosome availability in the cell, we implement this motif using RNA logic, where the actuator is a protein-encoding mRNA and the annihilator is a small RNA (sRNA) that binds it to form a rapidly degraded complex[53,54]. As it does not involve protein synthesis, this RNA-based implementation is neither affected nor disrupted by translational resource couplings that we seek to mitigate, whereas transcriptional couplings in bacteria are largely insignificant[6,14,55].

Besides the AIF motif's "actuator" and "annihilator" genes *act* and *anti*, the circuit comprises two more genes: the "sensor" *sens* and the "amplifier" *amp*. Ribosomal competition affects the level of the constitutively expressed transcription factor that is encoded by the sensor gene and regulates the annihilator sRNA's transcription. Actuation happens by changing concentrations of the transcripts competing for ribosomes. While the actuator mRNA sequesters some ribosomes, the protein encoded by it also regulates the transcription of an additional amplifier gene. This amplification is needed because heterologous mRNA concentration can significantly affect the ribosomal competition landscape only if it is large enough to be comparable to the

abundance of native transcripts. However, a substantial amount of actuator mRNAs is sequestered by the annihilator. Hence, achieving such a large concentration of heterologous transcripts by only expressing the actuator would require very high transcription rates. We therefore task another mRNA, which is not annihilated by any sRNA, with the bulk of the control action.

The variable of interest, whose fluctuations our controller aims to minimize, is the sensor protein's concentration $p_{sens}$, since the cell's resource availability is known to be captured by constitutive gene expression levels[56]. However, our model (with the simplifying assumptions outlined in the previous section) also allows to analytically retrieve the values of cell-wide variables that affect the expression of all genes in the cell. Namely, we can find the setpoint for the "resource competition denominator":

$$D = 1 + \frac{1}{1-\overline{\phi}} \sum_{q\,j\in\{a,r\}\cup X} m_j/k_j, \qquad (11)$$

In Eqs. (2)–(4), this value is the denominator of the effective protein synthesis rate constant, which adjusts the corresponding translation rate in view of competition from other transcripts. In a given culture medium and for a given steady-state value $\overline{p}_{sens}$, its setpoint value is given by Eq. (12). The meaning of the circuit's parameters found in this and other equations regarding our integral controller is illustrated in Fig. 6 and explained in Supplementary Tables 8 and 9.

$$\overline{D} = 1 + \frac{\overline{\lambda}\zeta}{(\overline{\lambda}+\beta_{sens})\overline{k}_{sens}^{NB}} \cdot \left( \frac{n_{sens}}{M} \cdot K_{sens} \cdot \frac{1-u}{u} \right)^{-1} \qquad (12)$$

Likewise, the steady-state growth rate maintained by the controller is given by:

$$\overline{\lambda} \approx \frac{\overline{\epsilon}^{NB}}{M} \cdot \overline{F}_r^{NB} c_r \alpha_r \cdot \frac{n_{sens}\overline{k}_{sens}^{NB}}{n_r \overline{k}_r^{NB}} \cdot \frac{K_{sens}}{c_{sens}\alpha_{sens}} \cdot \frac{1-u}{u} \qquad (13)$$

Besides setpoints for different variables, we can analytically estimate the controller's operation range—that is, the range of disturbance magnitudes which our design can negate to maintain a near-constant ribosome availability (Supplementary Note S4.4.2). If the cell is endowed with $L$ synthetic genes $\{x_1, x_2, ...x_L\}$ besides those of the controller, the disturbance caused by expressing them is mitigable only if:

$$\sum_{x_l \in \{x_1, x_2, ... x_L\}} \frac{\overline{F}_{x_l}^{NB} c_{x_l} \alpha_{x_l} \overline{\lambda}}{\overline{K}_{x_l}^{NB} (\overline{\lambda} + \beta_{x_l})} \leq \frac{c_{\text{sens}} \alpha_{\text{sens}} (1 - \overline{\phi}_q) M \overline{\lambda}}{K_{\text{sens}} n_{\text{sens}} \overline{k}_{\text{sens}}^{NB} (\overline{\lambda} + \beta_{\text{sens}})} \cdot \frac{1 - u}{u} \\ - \sum_{j \in \{a, r, \text{sens}\}} \frac{\overline{F}_j^{NB} c_j \alpha_j \overline{\lambda}}{\overline{k}_j^{NB} (\overline{\lambda} + \beta_j)}$$

(14)

To test the proposed controller's performance and the accuracy of our estimates, we used our cell model to simulate how the integral controller reacts to the appearance of an additional mRNA species competing for the shared ribosome pool (Fig. 7a), plotting the outcome in Fig. 7b–f. Figure 7b, c shows that the amplifier mRNA levels fall in response to a step disturbance, restoring the original extent of resource competition. Consequently, the adaptation error—that is, the difference between the variables' values before and after disturbance—decreased almost twofold compared to the open-loop case where the annihilator, the actuator, and the amplifier are not expressed. This reduction of adaptation error is observed for various disturbing gene concentrations within the calculated operation range (Fig. 7g) and stays consistent in presence of parameter uncertainty, stochasticity of gene expression, and time-variant disturbance (Supplementary Figs. 6 and 7). Besides the abundance of transcripts competing for ribosomes, resource availability depends on the culture medium's nutrient quality. As Fig. 8 shows, the controller successfully reduces the adaptation error upon the induction of a disturbing gene in different media, as well as when nutrient quality and the competing mRNA's abundance are varied together.

Nevertheless, the observed adaptation errors in Fig. 7d–f are nonzero, which can be explained by "leakiness". This phenomenon arises in AIF controllers when the actuator's and the annihilator's degradation and dilution rates are non-negligible compared to their rate of mutual elimination, which prevents the controlled variable from achieving RPA. Moreover, our derivations of the analytical estimates for $\overline{\lambda}$ and $\overline{D}$ neglect leakiness (Supplementary Note S4.4.2). Alongside the assumptions of constant $\epsilon$ and $F_r$ that we made in our derivations, this has likely contributed to the observed discrepancies between the numerically obtained and analytically estimated values for these variables. However, although leakiness can arbitrarily deteriorate the performance of AIF controllers, the observed adaptation errors remain relatively low, while changing our circuit's parameters can further improve the controller's performance. Figure 7g shows how increasing the amplifier gain $\chi$ (i.e., the maximum amplifier mRNA production rate) lowers the adaptation error even as the disturbance's magnitude rises. Increasing the actuator and the annihilator RNAs' transcription rate $\kappa$ has been shown to reduce the adaptation error of the controlled variable's mean value[53]—although in some cases this can lead to instability[57] (Supplementary Fig. 8).

A promising application for our controller is enforcing the assumption of modularity in synthetic biology, which is typically invalidated by resource couplings[12]. For instance, the shape of a gene's induction curve can be altered if competing synthetic genes are expressed in parallel[6]. However, our controller renders the induction curve more robust to different disturbances (Fig. 9). Notably, the inducible module, the disturbing gene, and the constitutive sensor gene in this case interact only via the shared resource pool and not directly. The inducible module's improved robustness to perturbations therefore demonstrates that our controlled variable—i.e., the

sensor gene's expression—indeed reflects resource availability at the level of the entire host cell, and that by keeping it constant our integral controller mitigates fluctuations in burden. Besides maintaining modularity, our controller could slow the loss of synthetic genes by engineered cells. A loss of heterologous gene expression to mutation normally means that the growth rate is less impaired by gene expression burden, so mutants outcompete the original engineered cells[58]. Conversely, as long as our controller itself remains functional, it keeps resource availability roughly constant regardless of whether any other synthetic genes are expressed. Without the reduction of translational burden, mutants lack the competitive advantage over the engineered cells.

Notably, our controller produces many transcripts that sequester ribosomes to release them when disturbances increase resource demand (Fig. 6), which burdens the cells expressing it. This reduction of resource availability and cell growth in exchange for control over cellular variables and robustness to perturbations is a common trade-off of growth rate controllers[59,60] and feedback circuits that mitigate fluctuations in resource demand[11,61]. Meanwhile, non-burdensome mitigation of cell-wide resource availability changes is currently restricted to feedforward architectures where the disturbing and the controller gene must be co-regulated[62]. Conversely, by sensing ribosome availability itself via the sensor gene's expression level, our feedback controller has no such requirement, mitigating perturbations irrespective of their cause.

## Discussion

The influence of resource competition on synthetic circuit performance, both through indirect interactions between different synthetic genes and through the burden imposed on the host cell, can be seen from several perspectives. One approach exclusively considers synthetic genes, applying effective rate frameworks to obtain a simple, easily interpretable model that predicts a circuit's behavior and the impact of changing its design parameters[6,9,12]. In this case, the interplay between heterologous gene expression and the host cell's state is either neglected[12] or abstracted and captured by simple phenomenological relations, which involve circuit-specific parameters without clear physiological meaning and do not mechanistically explain the effects of burden[18]. On the other hand, models of the entire bacterial cell reveal how resources are allocated between the host's own genes and those of the synthetic circuit[28,38]. However, highly complex cell models involving many variables and parameters obscure the key underlying processes and do not allow analytical derivations, requiring extensive numerical simulations to understand the effect of a given design choice. Meanwhile, excessive coarse-graining may leave out the interactions that are crucial for understanding bacterial resource allocation and designing gene circuits.

Our modeling framework combines the strengths of simplified resource competition analysis frameworks and cell models, balancing coarse-graining with accurate representation of gene expression and regulation. To this end, we apply an established resource-aware modeling framework, based on effective reaction rate constants[9,12], to a coarse-grained model which relates protein synthesis to cell growth via the finite proteome trade-off[28] and captures the dynamics of near-optimal resource allocation in bacteria by incorporating the principles of flux-parity regulation of gene expression[26]. The host cell model, spanning only six variables, can be easily augmented with ODEs mirroring Eqs. (1) and (2) for each heterologous gene to describe an arbitrary synthetic gene circuit. Resource- and host-aware simulations of a circuit's behavior can then be performed using our Matlab implementation of the model found at https://github.com/KSechkar/rc_e_coli[63], enabling rapid and cheap prototyping of resource-aware biomolecular controllers in silico[64]. Besides numerical analyses, our framework's simplicity allows to obtain a range of analytical relations between a synthetic circuit's design parameters and the host cell's

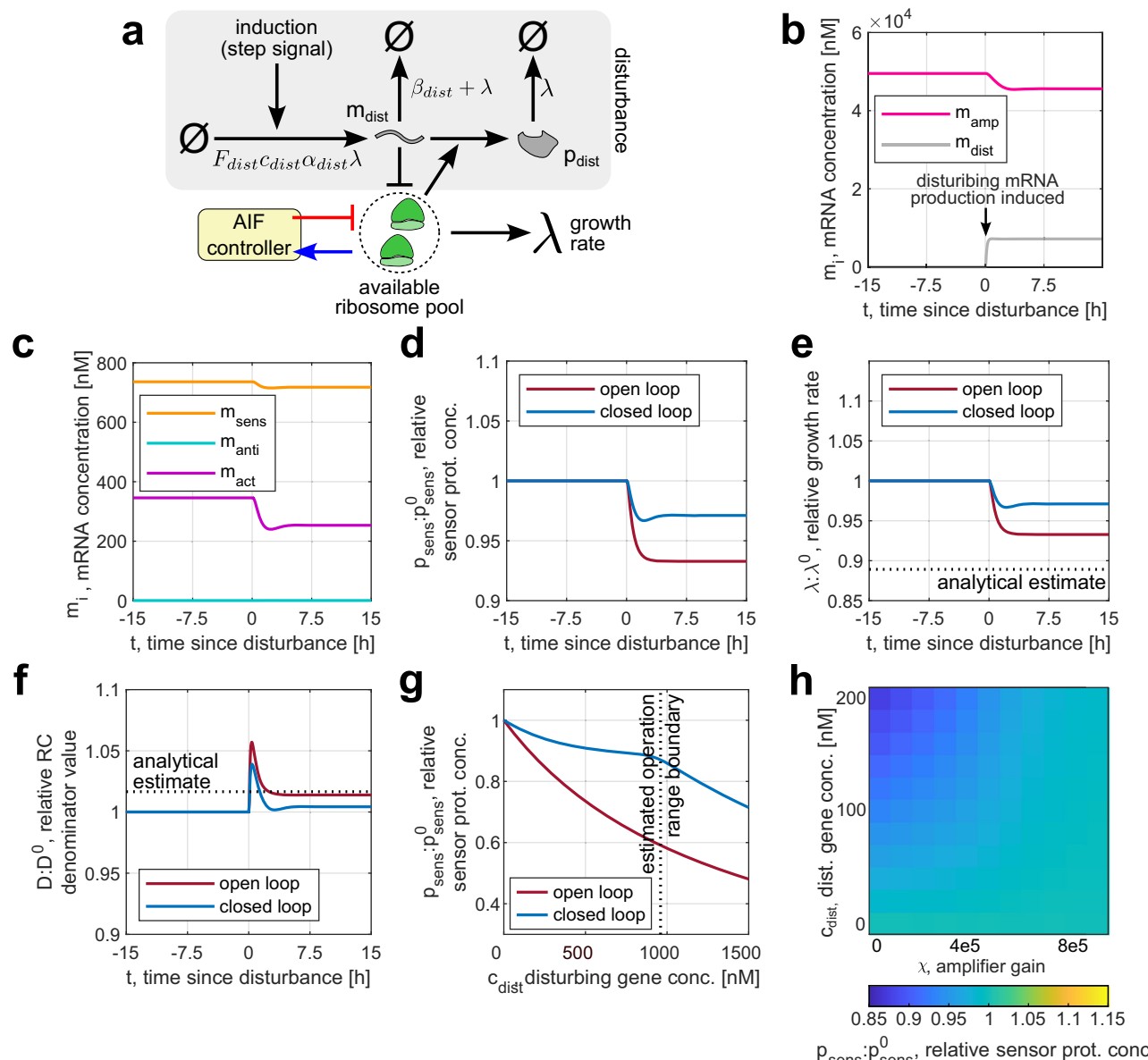

**Fig. 7 | Simulation of the integral controller's response to disturbance.**
**a** Resource availability in the cell is disturbed by inducing the expression of an additional disturbing synthetic gene *dist*. The resulting change in burden is counteracted by the controller. **b**, **c** Evolution of simulated mRNA concentrations over time upon the induction of a disturbing gene with the parameters given in Supplementary Table 9. **d**–**f** Simulated trajectories for the sensor protein's concentration $p_{sens}$ (the controlled variable), the cell's growth rate $\lambda$, and the resource competition denominator $D$ upon the induction of a disturbing gene. For comparison, the response of an open-loop system (i.e., in absence of the actuator, the annihilator, and the amplifier genes) is plotted on the same axes. All variables are plotted relative to their steady-state values. **g** Dependence of the steady-state value of $p_{sens}$, relative to that for an undisturbed cell, on the magnitude of disturbance—

that is, the concentration of the disturbing gene's DNA in the cell. Open-loop responses are plotted on the same axes for comparison. Outside of the calculated operation range, the controller no longer mitigates resource couplings, so the closed-loop value of $p_{sens}$ starts falling with increasing $c_{sens}$ at the same rate as the open-loop value. **h** The steady-state value of $p_{sens}$ maintained by the controller for various values of the amplifier gain $\chi$ and the disturbing gene's DNA concentration $c_{dist}$. The adaptation error increases as the disturbance rises but can be mitigated by increasing the amplifier gain. Unless specified otherwise, simulation parameters for all panels are given in Supplementary Table 9, and the ODEs used to simulate the controller are Supplementary Eqs. (121)–(129) in Supplementary Note S4.4.1. Source data are provided as a Source Data file.

growth rate, reproducing the empirical relations commonly used to capture burden in synthetic gene expression models[6,17–19]. Consequently, our model not only yields the forms of these relations which are rooted in physiological parameters, but also provides them with a unifying framework.

Comprising only a few variables, our model nonetheless escapes the major limitations of using very low-dimensional biomass-centric frameworks, such as the original flux-parity regulation model[26], in resource-aware circuit analysis. Namely, rather than considering molecules' fractions in the bacterial population's total biomass, our

model predicts their cellular concentrations, which are the quantities governing the biological interactions commonly leveraged in synthetic biology (e.g., transcription factor binding to DNA). Furthermore, as demonstrated by the different curve shapes in Fig. 5b, c, circuit design parameters' effect on the host cell is not directly reflected by the relationship between the cell growth rate and the synthetic protein mass fraction, but can be captured by our model. Unlike models that treat protein production as a single step[23,26,29], our framework also allows to model RNA-based regulation, which, by avoiding the most burdensome and resource competition-dependent step of

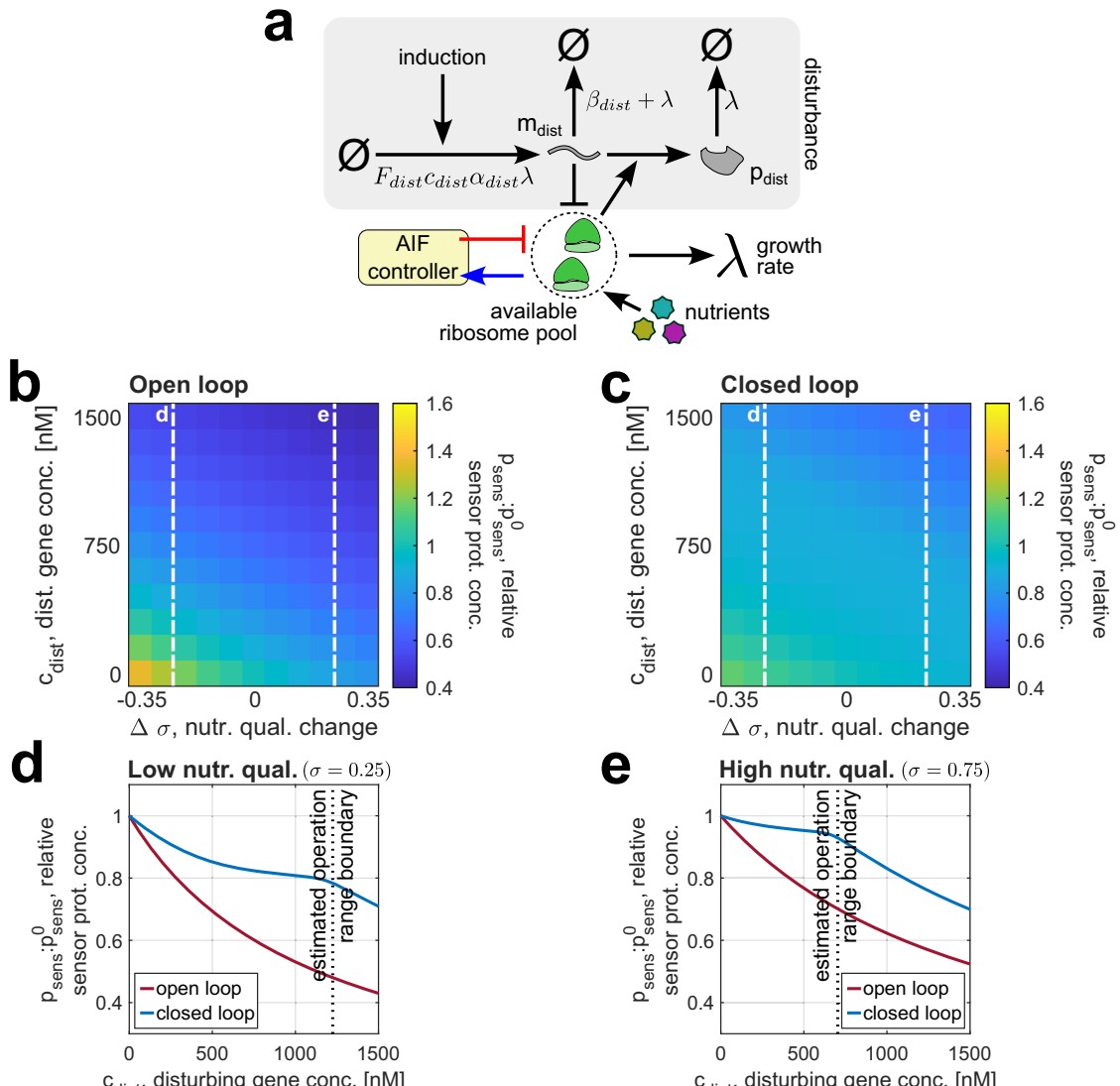

**Fig. 8 | Dependence of the controller's performance on the culture medium.**
**a** Resource availability in the cell can be perturbed both by introducing additional genes competing for ribosomes and changing the culture medium's nutrient quality, both of which our controller is designed to counteract. **b, c** Dependence of the steady-state value of $p_{sens}$ in an open- and closed-loop system, relative to that for an undisturbed cell in the default medium (i.e., $c_{dist} = 0$, $\sigma = 0.5$), on the disturbing gene's concentration and the change in the medium's nutrient quality. The dashed white lines mark the nutrient qualities considered in (**d**) and (**e**). **d, e** Dependence of the steady-state value of $p_{sens}$, relative to that for an undisturbed cell, on the disturbing gene's concentration for media with low ($\sigma = 0.25$, standing for $\lambda \approx 0.566 \, h^{-1}$ in the closed-loop case) and high ($\sigma = 0.75$, standing for $\lambda \approx 1.462 \, h^{-1}$ in the closed-loop case) nutrient qualities. Note that the undisturbed sensor protein concentration $p_{sens}^0$ is calculated for $c_{dist} = 0$ and the current nutrient quality−that is, $\sigma = 0.25$ or $\sigma = 0.75$, as opposed to (**b, c**), where it was defined for $\sigma = 0.5$ in all cases. Unless specified otherwise, the simulation parameters for all panels are given in Supplementary Table 9, and the ODEs used to simulate the controller are Supplementary Eqs.(121)−(129) in Supplementary Note S4.4.1. Source data are provided as a Source Data file.

translation[56], can be particularly useful in resource-aware circuit development.

We showcase our approach's advantages by designing and analyzing an integral controller that leverages RNA-based logic to mitigate fluctuations of the host cell's ribosome availability. The proposed controller acts at a cell-wide level, making our host-aware modeling framework more suited for its analysis than simpler resource competition models that neglect the cellular context. Our model enables numerical simulations of the controller. Furthermore, while the circuit infers resource availability via the proxy of the sensor gene's expression, the cell growth rate and extent of resource competition corresponding to the controller' setpoint, as well as our design's operation range, can be readily estimated using analytical relations.

Our model's reliability is supported by the concordance of its predictions with published experimental data. Although our model describes *E. coli* cells, the finite proteome trade-off is believed to be common for most growing bacterial cells[28], while several other bacteria, such as *S. coelicolor*, exhibit behavior consistent with flux-parity regulation of gene expression[26,65]. This makes our framework potentially applicable to other species upon re-parameterization. The model can also be extended to account for the aspects of cell physiology which are currently omitted but may be particularly relevant for certain applications. For instance, transcriptional resource couplings may become important in the cases of very high heterologous mRNA production. They can be captured by adopting the effective rate constant approach to simplify competitive RNA polymerase binding dynamics similarly to how we considered ribosome pool couplings[12].

In conclusion, our work presents a case for combining the insights from resource competition analysis with the appreciation of resource allocation in bacterial cells, which makes it an important step toward

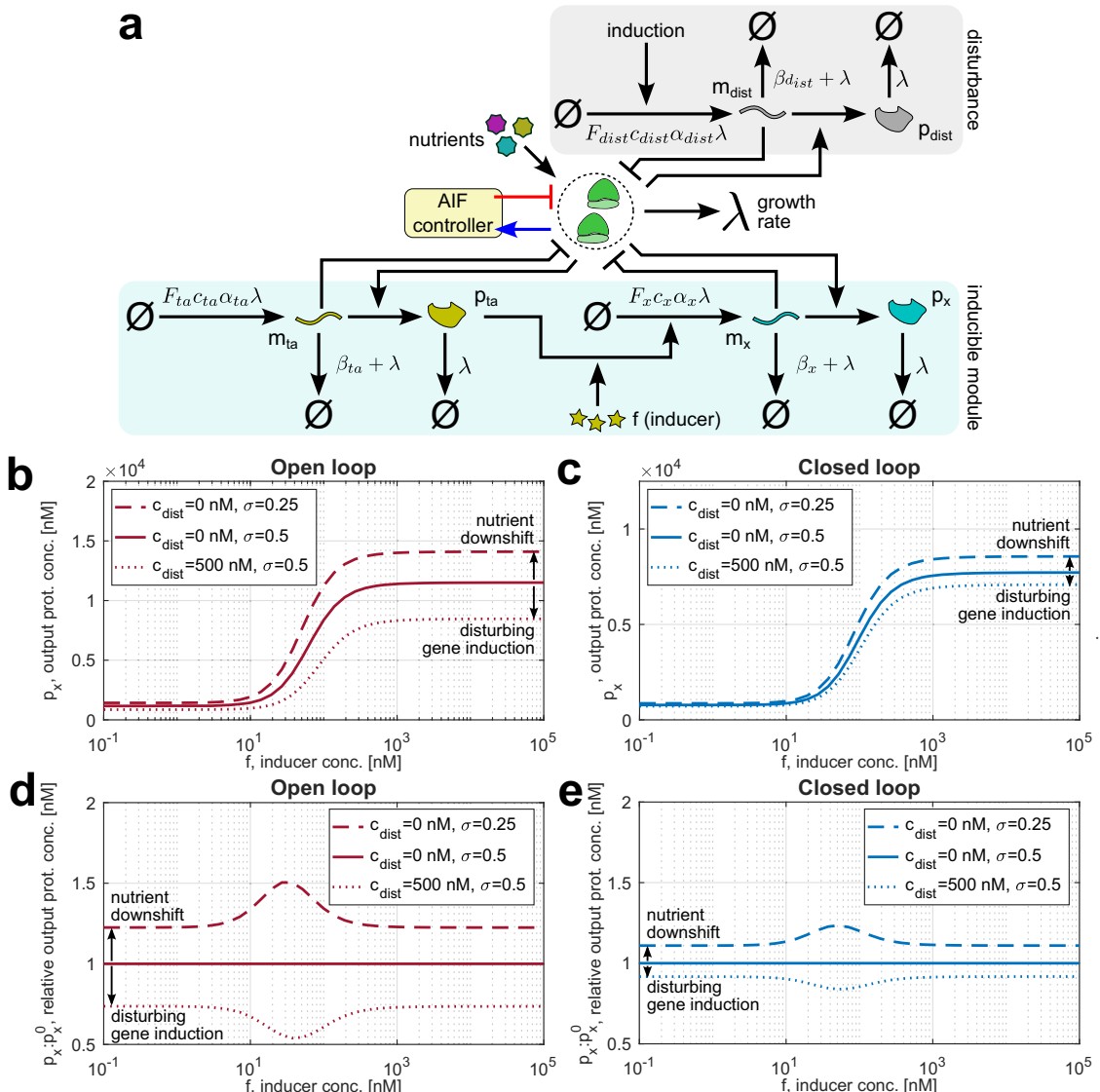

**Fig. 9 | The enforcement of modularity by our integral controller. a** Our integral controller mitigates the effect of competing synthetic genes and changes of the culture medium on the performance of an inducible genetic module, which consists of a transcription activation factor *ta* and the output gene *x* regulated by it. **b**, **c** The module of interest's induction curve (steady-state output protein concentrations plotted against inducer levels in the medium) in open- and closed-loop setups with and without different disturbances. **d**, **e** Open- and closed-loop steady-

state concentrations of the output protein $p_x$ relative to their values for a given inducer concentration *f* in the default culture medium and without disturbing gene expression. Unless specified otherwise, simulation parameters for all panels are given in Supplementary Table 9, and the ODEs used to simulate the controller are Supplementary Eqs. (121)–(134) in Supplementary Note S4.4.1. Source data are provided as a Source Data file.

easy design of reliable resource coupling-based controllers. Our model's predictive power can be further enhanced by considering more types of couplings via different resource pools and incorporating the results of future experiments characterizing cell physiology. By providing numerical and analytical tools for developing resource-aware circuits with an appreciation for their interactions with the host, we hope to encourage further exploration of bacterial resource-aware controller designs and open up avenues for holistic analysis of bio-circuit performance in the context of the host cell.

## Methods
### Model definition
Our mechanistic cell model is given by Eqs. (15)–(20), where $m_a$ and $m_r$ are respectively the concentrations of metabolic and ribosomal mRNAs, $p_a$ is the metabolic protein concentration, $R$ is the cell's total ribosome count, and $t^c$ and $t^u$ are the concentrations of aminoacylated

and uncharged tRNAs. The modeling assumptions giving rise to these equations are summarized in Table 1 and explained below:

$$\dot{m}_a = F_a c_a \alpha_a \lambda(\epsilon, B) - (\beta_a + \lambda(\epsilon, B)) m_a \tag{15}$$

$$\dot{m}_r = F_r(T) \cdot c_r \alpha_r \lambda(\epsilon, B) - (\beta_r + \lambda(\epsilon, B)) m_r \tag{16}$$

$$\dot{p}_a = \frac{\epsilon(t^c)}{n_a} \cdot \frac{m_a/k_a}{1 + \frac{1}{1 - \bar{\phi}_q} \sum_{j \in \{a,r\}} m_j/k_j} R - \lambda(\epsilon, B) \cdot p_a \tag{17}$$

$$\dot{R} = \frac{\epsilon(t^c)}{n_r} \cdot \frac{m_r/k_r}{1 + \frac{1}{1 - \bar{\phi}_q} \sum_{j \in \{a,r\}} m_j/k_j} R - \lambda(\epsilon, B) \cdot R \tag{18}$$

## Table 1 | Summary of modeling assumptions and equation terms affected by them

| Assumption | Equations | Relevant terms[a,b] |
|---|---|---|
| **mRNA synthesis** | | |
| Negligible transcriptional couplings | (15), (16), (21) | $F_i \cdot c_i \cdot \alpha_i \cdot \lambda$ |
| Constitutive metabolic gene expression | (15) | $F_a \equiv 1$ |
| mRNA synthesis rate proportional to cell growth rate | (15), (16), (21) | $F_i \cdot c_i \cdot \alpha_i \cdot \boldsymbol{\lambda}$ |
| **Protein synthesis** | | |
| Translational couplings captured by effective rate constants | (17), (18) | $\frac{\epsilon(t^c)}{n_i} \cdot \frac{\boldsymbol{m_i/k_i}}{\boldsymbol{1 + \frac{1}{1-\bar{\phi}_q} j \in \{a,r\} m_j/k_j}} R$ |
| Housekeeping protein mass fraction fixed | (17), (18), (22) | $\phi_q \equiv \bar{\phi}_q = 0.59$ |
| Housekeeping gene expression not modeled explicitly | (17), (18), (22) | $\frac{m_i/k_i}{1 + \frac{1}{1-\bar{\phi}_q} \sum_{j \in \{a,r\}} m_j/k_j}$ |
| **Degradation and dilution of species** | | |
| mRNA degradation rate comparable to cell growth rate | (15), (16) | $-(\lambda + \beta_i)m_i$ |
| Protein degradation rate negligible compared to cell growth rate | (17), (18) | $-\lambda p_i$ |
| tRNA degradation rate negligible compared to cell growth rate | (19), (20) | $-\lambda t^c, \; -\lambda t^u$ |
| **Cell growth rate regulation** | | |
| Cell growth rate maintains constant protein mass per unit of cell volume | (25) | $\lambda(\epsilon, B) = \frac{\epsilon B}{M}$ |
| **Translation elongation** | | |
| Translation elongation rate governed by Michaelis-Menten kinetics | (26) | $\epsilon(t^c) = \epsilon_{\max} \frac{t^c}{t^c + K_\epsilon}$ |
| **Protein precursor synthesis** | | |
| tRNA aminoacylation rate governed by Michaelis-Menten kinetics | (27) | $\nu(t^u, \sigma) = \boldsymbol{\nu_{max}} \sigma \frac{\boldsymbol{t^u}}{\boldsymbol{t^u + K_\nu}}$ |
| Culture medium's nutrient quality captured by the nutrient quality factor $\sigma$ | (27) | $\nu(t^u, \sigma) = \nu_{max} \boldsymbol{\sigma} \frac{t^u}{t^u + K_\nu}$ |
| **Flux-parity resource allocation** | | |
| ppGpp level reflects the ratio of charged and uncharged tRNA concentrations | (28) | $T = \frac{t^c}{t^u} \propto \frac{1}{[\text{ppGpp}]}$ |
| Ribosome synthesis regulated by ppGpp | (29) | $F_r(T) = \frac{T}{T+\tau}$ |
| tRNA and ribosome synthesis co-regulated | (30) | $\psi(T) = \psi_{max} \boldsymbol{F_r(T)}$ |

[a]Generic index $i$ means that the term pertains to both $a$ and $r$ genes.

[b]Where not all factors of a term reflect the assumption, the relevant part is given in bold.

$$\dot{t}^c = \nu(t^u, \sigma) \cdot p_a - \epsilon(t^c) \cdot B - \lambda(\epsilon, B) \cdot t^c \qquad (19)$$

$$\dot{t}^u = \psi(T) \cdot \lambda(\epsilon, B) - \nu(t^u, \sigma) \cdot p_a + \epsilon(t^c) \cdot B - \lambda(\epsilon, B) \cdot t^u \qquad (20)$$

**mRNA synthesis.** Gene expression and growth in bacteria are primarily affected by translational, rather than transcriptional, resource availability[6,14,55]. For simplicity, we therefore do not consider competition for transcriptional resources (e.g., RNA polymerases), so the rate of mRNA synthesis in Eqs. (15) and (16) is given by:

$$F_i \cdot c_i \cdot \alpha_i \cdot \lambda \qquad (21)$$

where $\alpha_i$ is the corresponding gene's promoter strength and $c_i$ is its DNA's concentration, which for the cell's native genes is assumed to be 1 nM as a convention (since the volume of an *E. coli* cell is around 1 μm³, this is equivalent to one gene copy per cell[12]). $F_i$ is the dimensionless transcription regulation function. For the ribosomal genes, $F_r$ captures the effects of ppGpp regulation and will be defined later in this section. The metabolic gene class as a whole is commonly treated as

constitutively expressed[19,21,22]. Likewise, the expression of aminoacyl-tRNA synthetases—which we assume to be part of the metabolic gene class, despite them sometimes not belonging to it in some other models[19,28,38]—exhibits little to no dependence on ppGpp levels[66]. Hence, we use $F_a \equiv 1$ across all culturing conditions. Finally, Eq. (21) includes the cell's growth rate $\lambda$, as RNA production rates across the bacterial genome increase linearly with the growth rate, presumably due to changes in the availability of the $\sigma^{70}$ factor, which is responsible for the transcription of most genes in exponentially growing bacteria[44,67]. Consequently, since the transcription rate is measured in nM of mRNA synthesized per hour, while the units of $c_i$ and $\lambda$ are respectively nM and h$^{-1}$, the promoter strength $\alpha_i$ is dimensionless. In Supplementary Note S2.3, we also discuss how the value of $\alpha_i$ captures the possibility of a single mRNA molecule being simultaneously translated by multiple ribosomes[12,68].

**Protein synthesis.** To define protein synthesis rates in Eqs. (17) and (18), we adopt an approach similar to that used in refs. 9, 12. Hence, in Supplementary Notes S1.2 and S1.3 we derive the effective translation rate constants:

$$k_i^{\text{eff}} = \frac{\epsilon}{n_i} \cdot \frac{1/k_i}{1 + \frac{1}{1-\bar{\phi}_q} \cdot \sum_{j \in \{a,r\}} m_j/k_j} \qquad (22)$$

where

$$k_i = \frac{k_i^- + \epsilon/n_i}{k_i^+} \qquad (23)$$

is the apparent mRNA-ribosome dissociation constant, determined by the binding and unbinding rates between the RBS and the ribosome ($k_i^+$ and $k_i^-$, respectively) and the rate at which the ribosome completes translation to slide off the mRNA. The latter can be obtained by dividing $\epsilon$, the translation elongation rate in amino acids per hour, by $n_i$, the number of amino acid residues in the protein encoded by the gene. Notably, owing to our assumption that the mass fraction of housekeeping proteins in the cell $\bar{\phi}_q \approx 0.59$ always remains constant[20,26], we avoid modeling the expression of housekeeping genes explicitly. Hence, Eq. (22) captures the housekeeping mRNAs' ribosome demand via the factor $\frac{1}{1-\bar{\phi}_q}$ (Supplementary Note S1.5).

**Dilution and degradation of species.** As for the removal of molecules, all species are diluted at a rate $\lambda$ due to the cell growing and dividing. The overwhelming majority of proteins are not actively degraded in the bacterial cell[69]. Hence, save for extremely unfavorable culture conditions that our model does not aim to describe, the total rate of protein breakdown in the cell is at most 0.02–0.025 h$^{-1}$, at least an order of magnitude smaller than the typical growth rates of 0.3–2 h$^{-1}$ considered in our study[70]. Protein degradation is therefore negligible compared to dilution. Conversely, mRNA degradation is widespread and happens at a high rate[49]. Hence, we include additional constant degradation terms $\beta_a$ and $\beta_r$ in Eqs. (15) and (16), which describe mRNA concentration dynamics. In contrast to mRNAs, the highly stable secondary and tertiary structure of tRNA molecules protect them from degradation both in and out of steady-state cell growth regimes, which makes the degradation rate of tRNA vanishingly small on the timescale of cell division[71,72]. Therefore, the species $t^c$ and $t^u$ in Eqs. (19) and (20) are removed by dilution only.

**Cell growth rate regulation.** The growth rate $\lambda$ is related to the rate of protein synthesis in the cell by the finite proteome cellular trade-off, identified by Weisse et al.[28]. While the shares of different protein classes in the cell's protein mass can vary, the total mass of proteins per unit of cell volume has been observed to be constant in the exponential growth phase[73]. Consequently, the overall rate of production of

all proteins must equal the total rate of protein mass dilution due to cell division. If $M$ is defined as the mass of proteins (in amino acid residues) in the average volume of the cell over the cell cycle and

$$B = \frac{\sum_{j\in\{a,r\}} m_j/k_j}{1 + \frac{1}{1-\phi_q}\sum_{j\in\{a,r\}} m_j/k_j} R \qquad (24)$$

is the total concentration of actively translating ribosomes (Supplementary Note S1.4), Supplementary Eq. (8) in Supplementary Note S1 shows that $\lambda$ can be found as the quotient of the overall protein synthesis rate (i.e., abundance of translating ribosomes times the rate of translation elongation) and the cell's protein mass, which yields Eq. (25):

$$\lambda(\epsilon,B) = \frac{\epsilon B}{M} \qquad (25)$$

**Translation elongation.** We now proceed to define the translation elongation rate $\epsilon$, which depends on $t^c$, the size of the pool of protein synthesis precursors (i.e., charged tRNAs)[22]. This relationship can be described with Michaelis-Menten kinetics[23] as shown in Eq. (26), where $\epsilon_{\max}$ is the maximum possible translation elongation rate[74] and $K_\epsilon$ is the half-saturation constant:

$$\epsilon(t^c) = \epsilon_{\max}\frac{t^c}{t^c+K_\epsilon} \qquad (26)$$

**Protein precursor synthesis.** To replenish the protein precursors consumed during translation, the uncharged tRNAs ($t^u$) are aminoacylated by metabolic proteins $p_a$. In line with our coarse-grained approach, we represent this process as a single reaction, whose rate per molecule of metabolic protein, which acts as the enzyme, is given by a Michaelis-Menten relation in Eq. (27)[26]. In this relation, $t^u$ is the substrate, $K_\nu$ is the corresponding half-saturation constant, and $\nu_{\max}$ is the maximum tRNA aminoacylation rate per enzyme. The nutrients are assumed to be present in excess—hence their concentration having no effect on the reaction rate—and the aminoacyl-tRNA yield per nutrient molecule (alias the medium's nutrient quality) is captured by a constant factor $0 \le \sigma \le 1$. Notably, some steps of converting the nutrients into protein precursors may in fact be catalyzed by housekeeping proteins. However, these enzymes' concentrations are unchanging across all culture conditions, so the rates of the reactions catalyzed by them can be factored into the constant parameters $\nu_{\max}$ and $\sigma$.

$$\nu(t^u,\sigma) = \nu_{\max}\sigma\frac{t^u}{t^u+K_\nu} \qquad (27)$$

**Flux-parity resource allocation.** Finally, we consider how the cell allocates resources between the expression of different genes, which is captured by the ribosomal gene transcription regulation function $F_r$. We implement the near-optimal regulation of bacterial gene expression by the ppGpp signaling pathway according to the recently proposed Flux-Parity Regulation Theory. Yielding reliable predictions of bacterial behaviors in both steady-state and dynamic scenarios, it postulates that the cell equates and maximizes the protein synthesis and tRNA aminoacylation fluxes, which allows it to achieve maximum growth rate[26]. The cellular concentration of ppGpp ($[ppGpp]$) therefore reflects the ratio of the aminoacylated and uncharged tRNA concentrations, although the exact mechanism enabling this is not yet entirely clear[31,75]. Consequently, instead of explicitly modeling the ppGpp concentration's dynamics, we describe it with a variable $T$ that follows the biochemically motivated phenomenological relation[26]:

$$T = \frac{t^c}{t^u} \propto \frac{1}{[ppGpp]} \qquad (28)$$

ppGpp represses ribosome synthesis[30,76], which can be captured by the Hill kinetics outlined in Eq. (29), where $\tau$ is the half-saturation constant:

$$F_r(T) = \frac{T}{T+\tau} \qquad (29)$$

Since tRNA genes are co-regulated with the ribosomal genes[77], $F_r(T)$ is likewise included in Eq. (30) for the rate of tRNA transcription $\psi$. This value is calculated per unit of growth rate because tRNA transcription is enabled by the same $\sigma^{70}$ factor as for most mRNAs[78], whose availability we assume to be growth-dependent[44]:

$$\psi(T) = \psi_{\max}F_r(T) = \psi_{\max}\frac{T}{T+\tau} \qquad (30)$$

**Numerical simulations**
Our Matlab model implementation, along with all other scripts and data used to obtain the results described here, can be found at https://github.com/KSechkar/rc_e_coli[63]. The deterministic simulations were run using Matlab R2022a's `ode15s` solver on a Dell OptiPlex 7000 PC with a 2.10 GHz 12th Gen Intel(R) Core(TM) i7-12700 processor and 16 GB RAM, running on Windows 11.

To investigate the integral controller's stochastic performance, we used a hybrid tau-leaping simulation algorithm[79]. Our host cell variables are coarse-grained, representing the average dynamics of multiple variables whose fluctuations can be expected to average out, so their dynamics were captured by the hybrid model's deterministic component. Meanwhile, heterologous gene expression dynamics were described by the stochastic model component[38]. The Matlab implementation of this simulation algorithm—the code for which can be found together with the rest of our scripts in this manuscript's GitHub repository—was run on a PC with a 3.20 GHz Intel(R) Xeon(R) w5-2455X processor and 125 GB RAM, running on Ubuntu 20.04.1. Matlab R2023b's `ode15s` solver was used to simulate the model's deterministic component, while the stochastic component was simulated using a tau-leap algorithm with a time step of $10^{-6}$ h. The details of our hybrid simulation method are provided in Supplementary Note S3.3.

All ODEs used in synthetic circuit simulations are provided in Supplementary Notes S3.1, S4.2.1, S4.3.1 and S4.4.1. Synthetic circuit parameters, displayed in Supplementary Tables 1, 4, 6, 7 and 9, were picked from the feasible ranges based on published literature[80–83] as outlined in Supplementary Note S4.1. Whenever a system's steady state was found by numerical simulation, this meant simulating the ODEs for 72 h by default, 48 h when calculating the likelihood function during parameter fitting (as a way of shortening the runtime of our code that needed to evaluate it many consecutive times), and 480 h when sweeping through different synthetic gene concentrations in Fig. 5 and Supplementary Fig. 4 due to the cell taking a long time to reach its steady state in presence of very high burden. The simulation was terminated prematurely if the changes in all variables over 12 h yielded less than $10^{-6}$ when squared and summed together.

**Parameter fitting**
Parameter fitting was used to estimate the maximum tRNA aminoacylation rate $\nu_{\max}$ and the ribosome-chloramphenicol binding rate $k_{cm}$ (required to model chloramphenicol's action on the cell as outlined in Supplementary Note S2.1), as well as $K_\epsilon$ and $K_\nu$, the Michaelis constants determining the translation elongation and tRNA aminoacylation rates. While the metabolic and ribosomal genes' promoter strengths $\alpha_a$ and $\alpha_r$ also had to be determined this way, we found that the optimality of the fits remained approximately unchanged over a wide range of values as long as the $\alpha_r{:}\alpha_a$ ratio was the same (Supplementary Fig. 2b). Therefore, we defined $\alpha_a$ using an order-of-magnitude estimate and fitted this ratio's value to experimental data. As we outline in

Supplementary Note S2.3, this then allowed us to obtain more accurate promoter strengths by matching our model's predictions to experimentally measured RNA production rates[84]. Moreover, since the mutual maximization of tRNA charging and protein synthesis fluxes is known to be achieved when $K_\epsilon \approx K_\nu$[26], we assumed them to be equal, reducing the number of parameter values to be fitted down to only four.

Experimental datapoints for fitting were taken from the study by Scott et al.[19], which measured steady-state growth rates and RNA:protein mass ratios of *E. coli* subjected to different concentrations of the translation-inhibiting drug chloramphenicol in various culture media. In order to convert RNA:protein mass ratios into ribosomal mass fractions, they were multiplied by a conversion factor of 0.4558[26]. Due to our model's lack of consideration of the cell's metabolic regulation mechanisms at very low growth rates, only the datapoints with $\lambda > 0.3$ were used for parameter fitting.

Similarly to Weisse et al.[28], who fitted their model's parameters to the same measurements and also used a constant scaling factor between 0 and 1 to quantify the yield of translation rate-defining precursor molecules per nutrient molecule, we estimated different media's nutrient qualities as six points equally spaced on the logarithmic scale between $\sigma = 0.08$ and $\sigma = 0.5$. The effect of chloramphenicol on the cell was modeled by Supplementary Eqs. (59)–(64), derived and explained in Supplementary Note S2.1. Briefly, the ODEs were obtained similarly to the original cell model, first defining a model that explicitly considers all reactions involved in competitive ribosome binding then simplifying the translation dynamics. However, as chloramphenicol binds and disables translating ribosomes, extra terms were introduced to the definitions of apparent mRNA-ribosome dissociation constants $\{k_i\}$, as well as to the ODE for $R$, redefined as the concentration of the cell's operational ribosomes. As the cell's measured ribosome content also includes inactivated ribosomes, their levels had to be considered and were denoted as a new variable $B_{cm}$, whose ODE was added to the model (Supplementary Note S2.1).

Fitting was performed using the DiffeRential Evolution Adaptive Metropolis (DREAM) algorithm, which is a variation of the Markov Chain Monte Carlo scheme for inferring parameters' probability distribution given a set of experimental observations. This involves following the trajectory of a Markov chain, whose states are sets of possible parameter values and whose stationary distribution is equal to the probability distribution of interest. Generally, such a chain is simulated by choosing a potential next set of parameter values according to some proposal distribution and then either accepting or rejecting it based on the probability of observing a known outcome if they are true. In the DREAM modification, the simulation's efficiency is increased by continuously adapting the proposal distribution based on the states considered in the past, as well as tracking several (in our case, 10) trajectories in parallel[85].

Priors for all parameters were defined as normal distributions. The mean was assumed to be 1 for $\alpha_r$:$\alpha_a$ and taken from literature for the other parameters[26,28,37]. The variances were assumed to comprise a quarter of the mean. The admissible intervals for all parameters were set between $\frac{1}{50}$ and 50 times the prior's mean (Supplementary Table 3). The DREAM simulation was run for 20,000 steps, using Matlab's Parallel Computing Toolbox v7.6 and the DREAM v2.4 Matlab package, part of the HydroSight v1.3.1 toolbox[86]. The mode of the fitted probability distributions provided the parameter values for our model. The parameters' starting values and other simulation details, as well as the details of parameter sensitivity analysis based on the fitting's outcome[87,88], can be found in Supplementary Note S2.2. The parameter fitting script is provided in this manuscript's GitHub repository[63].

### Reporting summary

Further information on research design is available in the Nature Portfolio Reporting Summary linked to this article.

## Data availability

The data generated in this study are provided in the GitHub repository https://github.com/KSechkar/rc_e_coli[63] under https://doi.org/10.5281/zenodo.10700011. The data used for parameter fitting and in Fig. 3a and Supplementary Fig. 2a are from the study of ref. 19 and are available as Supplementary Material at https://doi.org/10.1126/science.1192588. The data used in Figs. 3b–d and 5c and Supplementary Fig. 1 are from the study of ref. 26 and are available in the GitHub repository https://github.com/cremerlab/flux_parity[89] under 10.5281/zenodo.5893799. Source data are provided with this paper.

## Code availability

All code used to implement the model and obtain the figures is provided in the GitHub repository https://github.com/KSechkar/rc_e_coli[63] under https://doi.org/10.5281/zenodo.10700011.

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

## Acknowledgements

This work was funded by the RAE Chair in Emerging Technologies program by the Royal Academy of Engineering [RAEng CiET 1819\5 to G.B.S.].

## Author contributions

G.B.S., G.P., and K.S. conceived the study. K.S. developed the model and performed mathematical derivations. K.S. and H.S. developed the model's Matlab implementation. K.S. wrote the manuscript. G.P., G.B.S., and H.S. reviewed and edited the paper. G.P. and G.B.S. co-supervised the project. G.B.S. secured all funding.

## Competing interests

The authors declare no competing interests.
