## [Peer Review File · Nature Communications]

Reviewers' Comments:

Reviewer #1:

Remarks to the Author:

The manuscript titled "A coarse-grained bacterial cell model for resource-aware analysis and design of synthetic gene circuits" introduces a mathematical model based on ordinary differential equations to link metabolic burden and growth rate in bacterial cells. The topic is of central importance in synthetic biology, and the Introduction provides appropriate context and gives a detailed view of the current state of the topic. The paper has two main contributions: (1) the development of the coarse-grained bacterial cell model, and (2) the design of a proportional-integral biomolecular controller for mitigating gene expression burden. Unfortunately, both results suffer from major issues and limitations (as detailed below), questioning and likely compromising the novelty and impact of the work.

Major comments:

1) It is unclear how the coarse-grained mechanistic model (both with and without heterologous protein expression) represents an improvement over the one developed in ref [21] cited by the authors (pre-print available on bioRxiv since January 2022). Most of the assumptions and choices that the authors make can be found in the above paper. These include (i) a low-dimensional allocation model to describe the dynamic regulation of translational resources, representing the bottleneck in cellular growth and protein synthesis; (ii) division of proteome into ribosomal, metabolic, and housekeeping sectors; (iii) flux-parity regulation via a global regulator (ppGpp) capable of simultaneously measuring the turnover of charged- and uncharged-tRNA pools and routing protein synthesis; (iv) average metabolic rates or effective rate constants in lieu of mathematicizing the network's individual components; and (v) simple Michaelis-Menten type reactions. Importantly, the results in ref [21] offer a more comprehensive and well-developed framework and answers fundamental questions, such as "What biological mechanisms determine the allocation towards ribosomes in a particular environment and what criteria must be met for the allocation to ensure efficient growth?" and "How do cells coordinate their complex machinery to ensure optimal allocation?," among others. Additionally, the experimental validation in ref [21] is considerably more exhaustive, as only a handful of the panels presented in ref [21] are included in the manuscript. Furthermore, in these cases the predictions are often indistinguishable (as a result of the highly similar mathematical models and overall complexity), with the predictive power in ref [21] sometimes surpassing the one presented in this manuscript, for instance, regarding the translational elongation rate in the fast growth regime. Finally, whereas the current manuscript considers only steady state behavior for validation purposes, even transient changes are accurately captured in ref [21]. These include the quantitatively accurate description of multiple phenomena out of steady state (including nutrient upshifts and response to starvation) and under externally applied physiological perturbations (such as antibiotic stress or expression of synthetic genes). Taken together, the coarse-grained bacterial cell model that the authors have presented offers no apparent advantage over the one developed in [21], fundamentally limiting the novelty and impact of the paper.

2) Additionally, the presentation of the results makes it especially hard to follow the logic behind the equations, and several choices seem questionable/unjustified.

a) The introduction of the model should be reorganized, as numerous parameters and terms are introduced much later than where they are first used. For instance, $\psi(T)$ is introduced in equation (6), yet it is only defined 2.5 pages later.

b) Why include active degradation for mRNA, but not for protein and tRNA species?

c) Why is the production of protein and tRNA synthesis not proportional to growth rate, while mRNA synthesis is?

d) What does the $\epsilon \cdot B$ term represent in (5)-(6)?

e) Why does p_a appear in (13) as a single multiplier instead of as a Hill function, like t^c and t^u in (12)-(13), or p_{act} in equation (S85) in the Supplementary Information?

3) The insight from the winner-takes-all application example is unclear. Any one of the already existing models that account for competition for shared translational resources would yield similar qualitative behavior. Importantly, this could be achieved without including the impact of metabolic

burden on growth rate (for instance, when the authors explain the phenomenon, they do not refer to the role that growth rate modulation plays, since it is not essential). Without clearly demonstrating that the model developed here more accurately captures the dynamics from a quantitative perspective than already existing models using experimental data (as in ref [21] for the transient behavior), the results presented do not justify the validity of the modeling approach.

4) The analytical results rely on the assumption that “the translation elongation rate and ribosomal gene transcription regulation change very little over a wide range of heterologous gene expression rates.” Given its impact and non-trivial nature, this assertion should be validated experimentally (e.g., supported by prior experimental publications) instead of relying solely on the numerical simulations that the authors performed.

5) The proportional-integral biomolecular controller for mitigating gene expression burden is based on the antithetic motif, though here it is implemented using RNA instead of protein components to reduce translational burden. The proposed solution suffers from three major shortcomings:

a) For correct functioning, it seems that the controller sequesters a large amount of translational resources (primarily via the amplifier mRNA to create the corresponding protein), that it can then release once needed. That is, the controller imposes heavy burden on cells for the gratuitous expression of the amplifier protein, and then frees up some of these resources once they are needed for the synthesis of a target product. As a result, the mere presence of the controller represents a considerable fitness cost to cells, and this cost increases directly with the range of the controller (i.e., to regulate greater disturbances, it needs to sequester more resources beforehand in preparation). This is highly undesirable in the overwhelming majority of practical applications, as cells need to pay a constant high metabolic expense to prepare for the minimization of a potential future disturbance that may have limited cost.

b) The proposed controller regulates the term D instead of protein concentration directly, the quantity that is of primary focus in most practical applications. This is a considerable limitation: if cells face metabolic burden that affects their growth rate, the concentration of a target protein would change with the growth rate even when the controller is present (since it only regulates D , instead of the concentration itself).

c) The controller seems to suffer from a fundamental limitation when compared to the original antithetic motif. In case of the antithetic motif, the set-point value is a simple expression of two quantities controlled by the user, and it is robust to a large collection of disturbances in the regulated system and to inaccuracies about the underlying dynamical model. Conversely, the relationship between the quantity tuned by the user (u) and the setpoint \bar{D} is considerably more convoluted as defined in equation (29). As a result, it not only depends on numerous parameters of the biological system (most of which lump multiple physical processes together), but this expression also heavily relies on the model that the authors proposed earlier. This means that inaccuracies in the model and unavoidable perturbations in the cell (impacting any one of the lumped parameters) would compromise the performance of the controller. For instance, even assuming that the model perfectly captures the biophysical processes that govern cellular dynamics, if u is set in one context to ensure a given setpoint \bar{D} , if the context changes (e.g., as a result of internal or external perturbations changing the values of the parameters that appear in equation (29)), the setpoint would change accordingly (since u is unchanged), instead of remaining the same. This fundamentally questions the impact and applicability of the proposed controller module as it does not seem to be robust to common and ubiquitous perturbations.

6) To demonstrate the correct and robust functioning of the controller, the authors should perform an extensive set of numerical simulations. For instance, parameters of the coarse-grained model should be varied significantly (especially considering that the fit is not sensitive to most of these parameter values, see Supplementary Figure 1), and numerous disturbances should be considered, probing the effects of changes in the parameter values of the model as well as metabolic burden due to the expression of heterologous proteins. The scenarios should include not only steady-state but also time-varying burden, as well as stochastic analysis to demonstrate that noise does not compromise closed-loop performance.

7) Regarding the parameters of both the model and the controller, their values and units raise multiple issues.

a) Parameters of the model fit exhibit considerable “sloppiness” (Supplementary Figure 1), which

may suggest that the model could be further simplified to eliminate this. The authors should explore the manifold that the potential combinations are constrained on to reveal their interdependencies, and to further reduce the model order and/or the number of free parameters.

b) The typical range of all parameters should be sourced from experimental papers to ensure that the selected values represent realistic choices. Similarly, considering the simulated trajectories of the model (with and without the controller), the range of all species that appear in the model should also be evaluated against their typical ranges (this could also aid the selection of fitted parameters to ensure realistic parameterization).

c) The units of several parameters appear puzzling. For instance, the variable T is supposed to be dimensionless, whereas in Supplementary Table 2 it is indicated to be measured in nM. In the same table, the unit of ν is 1/h, whereas it should be nM/h. This issue persists throughout the Supplementary Information. The authors should carefully check all variables and parameters to ensure not only that their values are realistic, but also that their units are correct.

Minor comments:

- While the Abstract is well-written, it should be trimmed in length.
- The "Contents" on page 3 should be removed.
- There is a typo in (5): in the argument of ν the variable s should be σ .
- There are unresolved references in the Supplementary Information (e.g., on page 15).

Reviewer #2:

Remarks to the Author:

Overview of paper

This paper provides another contribution to understanding host-circuit interactions in synthetic biology. The authors' manuscript is clearly written and on the whole all equations are well described. The authors propose a small scale model of bacterial growth and gene expression with a core focus on ribosome and enzyme biosynthesis. The authors' model recapitulates well established experimental results in the field; accurately capturing the impact of translational inhibition, ribosomal mass fraction and elongation with growth rate. The authors show that their model captures the impact of host circuit interactions and demonstrate that it recapitulates the "winner-takes-all phenomenon" demonstrated experimentally by Xiao Jun-Tian and colleagues. The paper goes on to establish parameter regimes which enable quasi-steady state and other assumptions to be made which facilitates analytical solving of the model: collapsing the original 7 ODE host-circuit model to two algebraic equations. This analytical model is then used to establish how growth rate and heterologous protein production vary with translational burden. The authors propose a new synthetic gene circuit control system which maintains cell growth rate.

Major Comments

1) The authors model ppGpp regulation phenomenological though their parameter " T ". The authors' model only considers regulation of ribosomal production and tRNA production to be a function of T . ppGpp generally promotes transcription and can have significant transcription of amino acids as part of the "stringent response" (e.g. see <https://doi.org/10.1016/j.tim.2005.03.008>, <https://doi.org/10.1128/9781555816841.ch14>, and specific papers such as <https://doi.org/10.1073/pnas.0501170102>). While the authors' model (as is) fits experimental data in Figure 2, the authors should explain why they have not made F_a a function of T ? What impact do the authors think this would have on their model's findings?

2) In equation 13, the authors show that "metabolic proteins aminoacylate the uncharged tRNAs". Are the authors referring to amino acid producing enzymes or the aminoacyl-tRNA synthetases? In Weisse et al. 2015, the authors assume these accessory proteins form part of their ribosomal species as do Liao et al. in their 2017 paper. My understanding is that some aminoacyl-tRNA synthetases are co-transcribed from their amino acid synthesis operon – making them "enzymes" rather than "ribosomes". The authors should explain their rationale for this equation and highlight any differences from previous works to aid the reader in setting this work in context.

- 3) Please can the authors explain the biological rationale of their parameter σ (which governs efficiency of tRNA charging) and explain why they vary it between 0.08 and 0.5 in Figure 2.
- 4) Regarding the demonstration of their models in demonstrating the experimentally verified Winner-Takes-All phenomenon, please can the authors highlight which model has been used in the main text.
- 5) The power of the authors' analytical model is unclear to me. They define the protein production and growth rate in Eq. 23 and 24 which make reference to non-burdened values "NB". This enables an analysis to the unburdened cell. The authors demonstrate the use of this two-state analytical model in understanding cell growth and burden as well as identifying optimal production rates for a single heterologous protein. Can the authors demonstrate or discuss how their findings could be used to understand a more complex process? Robust analytical methods would be a powerful addition to the field but here it appears the system is only tractable as the problem is so simple.
- 6) In Eq 27 and the preceding sentence, please can the authors explain why the mRNA/ribosome dissociation constant is $k_{\text{poi}}^{\text{NB}}$ (i.e. related to the unburdened system?)
- 7) The authors' PI controller is interesting. Can the authors show the open loop dynamics in Figure 5b-f to aid readers understanding the controller mechanism? The controller acts to maintain growth rate (mitigates gene expression burden) without interfacing with ribosome biogenesis or needing new regulatory linkages. It looks to me that this system could be described as a controlled ribosomal sponge with p_{amp} sequestering ribosomes and then being tuned down by the PI controller through the act and sens modules. While this maintains the cell's burden (and so growth rate), this seems like a burdensome controller – i.e. the controller is required to sequester significant resources (which are then released for utilisation by the circuit genes dist). To increase the utility of their results (and facilitate future biological implementation) can the authors demonstrate the performance of their controller at various growth rates?
- 8) The authors demonstrate the controller functions well between 0 and 2 nM, but based on their assumption that 1 molecule per cell is 1 nM, this suggests the controller only functions for low burden proteins/circuits from chromosomally expressed genes. However the authors state in Table S7 that c_{dist} and c_{x} are 100 nM which would demonstrate a wider utility (with these higher values implying the system works for parameters equivalent to medium-to-high copy plasmids). Can the authors review how they present the performance of their controller to demonstrate its function over a range of biologically feasible α and c values (or demonstrate the function over $c^* = c \cdot \alpha$)?
- 9) Please can the authors give further details of their data fitting approach (e.g. cost function/loglikelihood functions) which are currently only available via the github repository.
- 10) In recent years there has been a proliferation of host-circuit interaction frameworks, including other developed by the authors, and a number are available as preprints. Many are cited by the authors, and I would urge the authors to reconsider their introduction to ensure none are missed. Most of these frameworks are based on similar concepts of ribosomal limitations as the authors. Please can the authors to explain more clearly the additional benefits of this new approach.

Minor Comments

- 1) In total the authors propose six coarse grained models of host circuit interactions. It would aid communication if the authors would make clear in their manuscript specifically which model has generated which results.
- 2) Figure 1 shows ppGpp explicitly and elsewhere the authors refer to “ppGpp regulation in our model”. However the authors’ inclusion of ppGpp is phenomenological (as opposed to mechanistic, as in e.g. Liao et al., 2017) – ideally the authors should make clear this is a phenomenological description given the wide readership of the journal.
- 3) The authors should detail in their methods how numerical steady states were identified; were simulations “just run for a long time” or was the derivative evaluated and time span of the simulation increased until the derivative was small?
- 4) Page 9. Should “s” in Equation 5 should be sigma?
- 5) Page 13. Reference to 2d should be 2e?
- 6) When discussing “our model achieves near-optimal steady state growth rates”, I am not sure if the authors are claiming this as a new finding and they could set this work in context more clearly by making reference to other works (e.g. by Hidde de Jong and Terrance Hwa which show theoretically and experimentally how ppGpp control of bacterial resource allocation seems to be the solution to an optimal control problem – the authors already cite some of these works but not in this section)
- 7) The authors could highlight the importance of their work clearer by comparing their lumped factor zeta with the J metric of Santos-Navarro et al. (ref 24) and the S and Q metrics of McBride and Del Vecchio (ref 6).

A coarse-grained bacterial cell model for resource-aware analysis and design of synthetic gene circuits: Response to Peer Review Comments

November 3, 2023

Foreword

We would like to thank the editor and reviewers for their time and effort dedicated to providing feedback for our manuscript. In this document, we provide point-by-point responses to the questions and comments that have been made. In our manuscript of the article, the amendments made in response to these comments are displayed in green.

1 Reviewer 1

1.1 Major Comments

1.1.1 Major Comment 1

It is unclear how the coarse-grained mechanistic model (both with and without heterologous protein expression) represents an improvement over the one developed in ref [21] cited by the authors (pre-print available on bioRxiv since January 2022). Most of the assumptions and choices that the authors make can be found in the above paper. These include (i) a low-dimensional allocation model to describe the dynamic regulation of translational resources, representing the bottleneck in cellular growth and protein synthesis; (ii) division of proteome into ribosomal, metabolic, and housekeeping sectors; (iii) flux-parity regulation via a global regulator (ppGpp) capable of simultaneously

measuring the turnover of charged- and uncharged-tRNA pools and routing protein synthesis; (iv) average metabolic rates or effective rate constants in lieu of mathematicizing the network’s individual components; and (v) simple Michaelis-Menten type reactions. Importantly, the results in ref [21] offer a more comprehensive and well-developed framework and answers fundamental questions, such as “What biological mechanisms determine the allocation towards ribosomes in a particular environment and what criteria must be met for the allocation to ensure efficient growth?” and “How do cells coordinate their complex machinery to ensure optimal allocation?,” among others. Additionally, the experimental validation in ref [21] is considerably more exhaustive, as only a handful of the panels presented in ref [21] are included in the manuscript. Furthermore, in these cases the predictions are often indistinguishable (as a result of the highly similar mathematical models and overall complexity), with the predictive power in ref [21] sometimes surpassing the one presented in this manuscript, for instance, regarding the translational elongation rate in the fast growth regime. Finally, whereas the current manuscript considers only steady state behavior for validation purposes, even transient changes are accurately captured in ref [21]. These include the quantitatively accurate description of multiple phenomena out of steady state (including nutrient upshifts and response to starvation) and under externally applied physiological perturbations (such as antibiotic stress or expression of synthetic genes). Taken together, the coarse-grained bacterial cell model that the authors have presented offers no apparent advantage over the one developed in [21], fundamentally limiting the novelty and impact of the paper.

Response: We would like to thank the Reviewer for highlighting the need to outline more clearly the particular aims of our model and why it is uniquely suited for achieving them. Our model is primarily intended to be used in synthetic biology, rather than systems biology, applications – namely, resource-aware design of genetic circuits. Therefore, the latest insights from systems biology put forward in prior publications (among which is the one listed as ref [21] in our original manuscript and ref [26] in the revised version), were put together with the aim to provide a modeling framework to facilitate gene circuit development.

The development of resource-aware cell models for different applications commonly involves combining existing insights from published literature in a way that best serves a given goal. For example, apart from the particular definitions of gene expression regulation functions, most features of ref [21] mentioned by the reviewer – namely, the assumption of simple Michaelis-Menten kinetics for cellular processes; the division of the cell’s proteome into metabolic, ribosomal, and

housekeeping classes; the treatment of translational resources as a key determinant of cell growth; the coarse-grained view of cellular processes using average metabolic rates or effective rate constants – are all present in previously published cell models such as Liao et al. 2017 and Santos-Navarro et al. 2021. Nevertheless, by putting the focus on the implementation of flux-parity regulation of resource allocation, ref [21] presents a powerful and comprehensive tool aimed at understanding which mechanisms govern the allocation of translational resources in bacteria and how this allows to optimize the cell’s metabolism.

From the same point of view of a cell model’s relevance for a particular application, several important biocircuit design needs are not met by the framework proposed in ref [21], as it has been developed with primarily systems biology goals in mind. Specifically, it does not consider gene expression at the level of a single cell, and rather focuses on the overall biomass of all cells in the culture. It also does not model transcription and translation as separate processes – instead, the rate of a given protein’s synthesis is considered via its mass fraction in the cell times the rate of biomass accumulation. This biomass-centric view offers no direct way to relate gene expression to the design parameters of synthetic gene circuits, such as promoter strengths, gene copy numbers, and RBS strengths. Important implications of certain synthetic circuit design choices can therefore be missed – for example, the experimentally observed Hill dependence between synthetic gene expression and cell growth (Figure 5c in our manuscript) would be impossible to reproduce by considering protein mass fractions only, since their relationship with cell growth is different, as is illustrated in Figure 5d. The biological interactions underlying synthetic circuit behavior (e.g., transcription factor-promoter binding) also often depend on protein abundance *per cell*, which further complicates the use of the model from ref [21] in forecasting circuit performance.

Conversely, the model we propose focuses on enabling resource-aware gene circuit design, relating the rates of genes’ expression to their design parameters. To this end, we leverage the features of different previously defined models which, as we explicitly acknowledge in our manuscript (lines 555–560), indeed include the flux-parity allocation theory as a low-dimensional yet accurate representation of near-optimal gene regulation mechanisms in bacteria. Besides offering a clear insight into the influence of different parameters on circuit performance, our model allows to reproduce both the linear and Hill relationships in Figure 5d and 5c respectively, showing how they can in fact have a common mechanistic basis.

Without singling ref [21] out, we have significantly rewritten the Introduction and Discussion

in our manuscript to highlight why we believe our model is a more effective tool for resource- and host-aware design of biomolecular controllers than the many previously proposed cell modeling frameworks. Balancing realism and detailedness with simplicity and interpretability, with only a handful of variables our framework captures near-optimal cellular resource allocation while allowing to model synthetic gene circuits that involve sundry regulatory mechanisms, such as transcriptional regulation by proteins, indirect interactions via resource competition couplings, and RNA-based integral feedback motifs. For instance, as we now highlight in lines 445–451 of the main text’s Discussion, this allows us to model a resource-aware antithetic integral feedback controller. Due to its negligible effect on ribosome availability, RNA-based integral feedback has served as a crucial part of multiple resource-aware biomolecular controllers (e.g., see Huang et al. 2018 and Boo et al. 2023); however, such circuits heretofore have not been studied using a host-aware framework that explicitly consider the bacterial cell’s metabolism, hence the limited power of modeling predictions in prior studies. In addition to enabling realistic numerical experiments, the model is also sufficiently simple to allow the derivation of analytical relations that provide useful insights into the behavior of gene circuits, such as the formulae for our integral controller’s operation range and setpoint cell growth rate and resource availability given by Equations (12)–(14) in the main text.

The Reviewer also mentions ref [21]’s ability to predict the cell state during out-of-steady-state nutrient upshifts and starvation, as well as its superior accuracy for predicting fast growing cells’ translation elongation rates when compared to experimental measurements. To provide an example of how our model can capture dynamic changes in resource allocation, we have added the new Supplementary Note S1.6, which extends our framework to simulate the cell’s behavior during a nutrient upshift and compares the numerical outcomes with experimental measurements out of steady state. Meanwhile, the comparison of different models’ predictive power is not straightforward. Both in this work and ref [21], as well as many other articles to date (e.g., Weisse et al. 2015 or Santos-Navarro et al. 2021), the evaluation of model predictions against experimental datapoints that it had not been fitted to is rather visual and does not provide quantitative metrics to assess the error, focusing on similar trends being observed for data and model forecasts. Indeed, quantitative comparison of errors between different models is of limited value when the modeling frameworks’ goals are different – such as resource-aware circuit design and the establishment of the fundamental principles of cellular resource allocation for our article and ref [21]. Moreover, even when two models’ aims are aligned, a somewhat less realistic framework may still prove to

be a useful design tool due to lower computational complexity, greater simplicity, or superior user-friendliness. For instance, likewise to our modeling framework, the cell model defined by Liao et al. 2017 was developed to study synthetic gene circuit behavior in view of host-circuit interactions and bacterial resource allocation. However, its higher dimensionality does not allow to obtain the analytical relations providing insight into biocircuit performance, which we derive using our model (see lines 231–271 and 348–365).

1.1.2 Major Comment 2

Additionally, the presentation of the results makes it especially hard to follow the logic behind the equations, and several choices seem questionable/unjustified.

- *The introduction of the model should be reorganized, as numerous parameters and terms are introduced much later than where they are first used . For instance, $\psi(T)$ is introduced in equation (6), yet it is only defined 2.5 pages later.*

Response: We thank the Reviewer for the constructive feedback. To facilitate comprehension, we now introduce our modeling framework graphically in Figure 2, assisted by a model schematic and a term-by-term visual explanation of the ordinary differential equations. The explanation and discussion of our modeling choices has been relocated to the Methods section so as to avoid breaking up the narrative.

- *Why include active degradation for mRNA, but not for protein and tRNA species?*

Response: Thank you for raising this point. Briefly, while the half-life of mRNA is much smaller than the cell’s division time, tRNA molecules belong to the so-called “stable RNA” class, which makes their rate of degradation very small compared to that of dilution due to cell growth and thus enables us to neglect tRNA active degradation in our model. Similarly, the active degradation rates of the majority of cellular proteins have been found negligible compared to dilution due to bacterial cell growth. We have now added this argument to the Methods section (lines 517–521 and 523–527), providing additional citations to support our reasoning, such as Pine, 1970, Chaitanya, 2002, Li and Deutscher, 2008, and Prossliner et al., 2023.

- *Why is the production of protein and tRNA synthesis not proportional to growth rate, while mRNA synthesis is?*

Response: We would like to thank the Reviewer for bringing up an important point about the dependence of tRNA synthesis rates on cell growth. Indeed, the expression of tRNA genes is enabled by the σ^{70} factor, the changes in whose availability are suggested to cause the scaling of mRNA synthesis rates with cell growth rates (see Balakrishnan et al. 2022). To make tRNA production depend on cell growth, our model’s definition of tRNA synthesis rate ψ has been amended (see Supplementary Tables S1 and S2 and the main text’s Equation (20)). The updated model’s parameters were refitted to data following the same procedure as before, and all figures were recreated using the new modeling framework. As it can be seen in the revised manuscript, our findings have not been qualitatively altered by these updated modeling assumptions. Moreover, as we discuss in the newly added Supplementary Note S3.3, the scaling of the tRNA synthesis rate with cell growth lends more credibility to the assumptions we use in our analytical derivations that the Reviewer has commented on in their Major Comment 4.

As for the link between protein production rates and cell growth, understanding it is a key aim of cellular resource allocation modeling. Rather than define a simple empirical relation as we do for RNA synthesis rates, we therefore capture it mechanistically by considering the finite proteome cellular trade-off in Equation (25), the dependence of ribosome availability on ppGpp regulation in Equation (29), and competition for resources in Equations (2), (17), and (18). To highlight the interdependence of cell growth and protein synthesis, lines 528–529 and 536–538 in the main text’s Methods have now been rewritten.

- *What does the ϵB term represent in (5)-(6)?*

Response: ϵB is the total rate of consumption of aminoacyl-tRNAs by all translating ribosomes (whose concentration is denoted by B in our model) in the cell. We hope that this has now been made clearer by the new Figure 2 and the reorganized explanation of our modeling framework in Methods.

- *Why does p_a appear in (13) as a single multiplier instead of as a Hill function, like t^c and t^u in (12)-(13), or p_{act} in equation (S85) in the Supplementary Information?*

Response: As we mention in the manuscript (lines 544–546), Equation (27) – formerly (13) – represents a standard case of Michaelis-Menten enzyme kinetics. The tRNAs that are being aminoacylated (t^u) are the substrate, hence them appearing in a Hill function. Meanwhile, the metabolic protein p_a is the enzyme, to whose total concentration the reaction rate is related linearly.

1.1.3 Major Comment 3

The insight from the winner-takes-all application example is unclear. Any one of the already existing models that account for competition for shared translational resources would yield similar qualitative behavior. Importantly, this could be achieved without including the impact of metabolic burden on growth rate (for instance, when the authors explain the phenomenon, they do not refer to the role that growth rate modulation plays, since it is not essential). Without clearly demonstrating that the model developed here more accurately captures the dynamics from a quantitative perspective than already existing models using experimental data (as in ref [21] for the transient behavior), the results presented do not justify the validity of the modeling approach.

Response: Rather than an attempt to quantitatively predict a system’s dynamic behavior, the “winner-takes-all” case is an example of how to modify our model to represent the expression of a synthetic gene circuit by the host cell. We then show that our implementation of the model can indeed capture the changes in synthetic circuit performance induced by resource competition. Primarily using it just as a showcase application for our model, we therefore do not claim to produce any novel insight on the “winner-takes-all” phenomenon compared to prior studies. As such, this showcase is meant to show the applicability of our model to previously considered cases, whereas novel directions and circuit designs are discussed in the later subsections of the manuscript’s Results (starting from line 231 and up to line 410).

We would like to thank the Reviewer for suggesting that other case-studies might be helpful to show our model’s broad applicability beyond the “winner-takes-all” phenomenon that is not directly enabled by changes in the host cell’s growth rate, which are a key factor that our model seeks to capture. We have therefore added a second showcase application for our model (lines 222–229 and new Figure 4c–e), revealing how it can be used to reproduce the phenomenon of emergent bistability that was observed experimentally in Tan et al. 2009. In this case, the slowdown of cell growth due to synthetic gene expression introduces an additional feedback loop that can confer bistability to a non-cooperatively self-activating gene, which would otherwise always have a single stable equilibrium. Reproducing both the “winner-takes-all” phenomenon, which arises from resource competition among different synthetic genes, and emergent bistability, enabled by the effect of burden on the host cell, we demonstrate that our model successfully captures resource competition phenomena of different nature.

Evaluating model predictions against experimentally measured properties of the bacterial cell,

likewise to what is done in ref [21] or for our model in Figure 3, is rather straightforward, since the same quantities, such as cell growth rate or ribosome content, are measured in different experiments for generally identical cells (although some variation between strains must be admitted). Conversely, quantitatively comparing predicted synthetic circuit behavior with published experimental data is more complicated. A synthetic gene circuit can be implemented using a wide variety of possible genetic parts and in different culture conditions, which are not always fully characterized in the published studies that describe resource competition phenomena. Additional unknowns are introduced by the common lack of absolute protein quantification. Indeed, reliable protocols for gauging protein abundance have started to emerge only recently (see Csibra and Stan 2022), hence most circuit experiments to date only measuring the reporter protein’s fluorescence in arbitrary units. Therefore, most cell modeling studies that rely on published experimental measurements (see, for instance, Liao et al. 2017 and Weisse et al. 2015) compare model predictions for the host cell’s state with experimental data, but make predominantly qualitative forecasts for circuit behavior, employing generic circuit parameter values. Following common practice in resource-aware cell modeling, we therefore chose not to plot experimental data in Figure 4, providing qualitative rather than quantitative predictions for the “winner-takes-all” and emergent bistability phenomena.

1.1.4 Major Comment 4

The analytical results rely on the assumption that “the translation elongation rate and ribosomal gene transcription regulation change very little over a wide range of heterologous gene expression rates”. Given its impact and non-trivial nature, this assertion should be validated experimentally (e.g., supported by prior experimental publications) instead of relying solely on the numerical simulations that the authors performed.

Response: Due to the difficulty of measuring translation elongation rates directly, no studies that we know of have investigated their dependence on synthetic gene expression burden. However, when the growth-dependence of tRNA transcription rates is taken into account as suggested by the Reviewer in Major Comment 2, the near burden-independence of the translation elongation rates and the ribosomal gene transcription regulation function mathematically follows from our model’s assumptions. A derivation for this is provided in the newly added Supplementary Note S3.3.

We would also like to point out that in Figure 5 the analytical predictions obtained with the help of our assumptions (dashed lines) are at all times very close to the results of straightforward

numerical simulations, vindicating our choice to simplify the model in our analytical derivations.

An acknowledgment of the lack of experimental data clearly in favor or against our assumption of constant translation elongation rate and ribosomal gene transcription regulation function has been added to the Main text’s lines 235–237 and lines 390–392 in the Supplementary Information.

1.1.5 Major Comment 5

The proportional-integral biomolecular controller for mitigating gene expression burden is based on the antithetic motif, though here it is implemented using RNA instead of protein components to reduce translational burden. The proposed solution suffers from three major shortcomings:

- *For correct functioning, it seems that the controller sequesters a large amount of translational resources (primarily via the amplifier mRNA to create the corresponding protein), that it can then release once needed. That is, the controller imposes heavy burden on cells for the gratuitous expression of the amplifier protein, and then frees up some of these resources once they are needed for the synthesis of a target product. As a result, the mere presence of the controller represents a considerable fitness cost to cells, and this cost increases directly with the range of the controller (i.e., to regulate greater disturbances, it needs to sequester more resources beforehand in preparation). This is highly undesirable in the overwhelming majority of practical applications, as cells need to pay a constant high metabolic expense to prepare for the minimization of a potential future disturbance that may have limited cost.*

Response: While our controller is rather burdensome, reduced resource availability is a common feature of controllers that mitigate competition for a limited pool of resources. Indeed, negative feedback is a core motif of our and other designs proposed to this end, e.g., Darlington et al. 2018 for orthogonal ribosomes or Huang et al. 2021 for dCas9 CRISPR moieties. The greater is the value of a negatively autoregulated variable, the more strongly negative feedback counters its further increases, so it is inherently prevented from reaching very high values. When resource availability becomes such a controlled variable, the controller therefore generally lowers its value in absence of disturbance. However, this pays off when negative feedback mitigates perturbations and renders a variable’s value robust to disturbances.

Despite also lowering resource availability for the sake of making it more robust to perturbations, it is true that the provided examples of existing negative feedback controllers do not necessarily exert a very high burden on the cell, since they only regulate the resources used by

synthetic gene circuitry and not the host’s native genes. However, for the same reason they cannot mitigate cell-wide changes in resource availability – such as those caused by alterations of the medium’s nutrient quality. Our controller, conversely, readily counters them as shown in Figures 8 and 9, newly added to address the third subpoint of this comment to demonstrate our controller’s ability to maintain resource availability near the setpoint value in sundry conditions, making synthetic circuit performance robust to perturbations of different nature.

Moreover, studying a burdensome circuit like ours generally offers a particularly relevant example of our model’s usefulness. Indeed, high burden can be understood as strong influence of synthetic gene expression on the host cell’s state, which in turn can affect a circuit’s performance by changing the cell’s growth rate and resource availability. Thus, understanding and predicting these complex interactions is a prime motivation for the use and development of coarse-grained resource-aware cell models. Our controller, which burdens the cell non-negligibly, therefore presented a very suitable case study of deploying the proposed modeling framework.

- *The proposed controller regulates the term D instead of protein concentration directly, the quantity that is of primary focus in most practical applications. This is a considerable limitation: if cells face metabolic burden that affects their growth rate, the concentration of a target protein would change with the growth rate even when the controller is present (since it only regulates D , instead of the concentration itself).*

Response: Thank you for highlighting the limitations of presenting D as the controlled variable in our design. Following your advice, we have rewritten our description of the integral controller (lines 348–350), redefining our controlled variable to be the concentration of the sensor protein p_{sens} .

At the same time, it can be seen that our controller does not merely keep the value of p_{sens} unchanging. Since constitutive protein expression reflects the cell’s resource availability (see Ceroni et al. 2015), maintaining a constant value of p_{sens} still results in making the overall extent of resource competition in the cell robust to perturbations. Namely, as we show in Figure 7 and Supplementary Figures S5–S7, the controller significantly reduces the adaptation errors upon perturbation both for the host cell’s growth rate λ and the extent of resource competition in the cell quantified by D . Moreover, our circuit can confer robustness to perturbations to an inducible genetic module, which does not engage in any direct interactions with the sensor gene (Figure 9

and lines 395–403). This further highlights the cell-wide nature of resource coupling mitigation enabled by our controller.

- *The controller seems to suffer from a fundamental limitation when compared to the original antithetic motif. In case of the antithetic motif, the set-point value is a simple expression of two quantities controlled by the user, and it is robust to a large collection of disturbances in the regulated system and to inaccuracies about the underlying dynamical model. Conversely, the relationship between the quantity tuned by the user (u) and the setpoint \bar{D} is considerably more convoluted as defined in equation (29). As a result, it not only depends on numerous parameters of the biological system (most of which lump multiple physical processes together), but this expression also heavily relies on the model that the authors proposed earlier. This means that inaccuracies in the model and unavoidable perturbations in the cell (impacting any one of the lumped parameters) would compromise the performance of the controller. For instance, even assuming that the model perfectly captures the biophysical processes that govern cellular dynamics, if u is set in one context to ensure a given setpoint \bar{D} , if the context changes (e.g., as a result of internal or external perturbations changing the values of the parameters that appear in equation (29)), the setpoint would change accordingly (since u is unchanged), instead of remaining the same. This fundamentally questions the impact and applicability of the proposed controller module as it does not seem to be robust to common and ubiquitous perturbations.*

Response: As mentioned in the previous part of our answer to Reviewer 1’s Major Comment 5, we now address this issue by considering p_{sens} as the primary controlled variable. Its setpoint value is independent of perturbations and calculated exclusively from u and the half-saturation constant for the transcription factor-annihilator promoter binding – that is, K_{sens} – according to the Supplementary Information’s Equation (S133). While this still leaves the setpoint vulnerable to changes in the actuator and annihilator genes’ parameters, the same can be said about the original AIF motif in Briat et al. 2016, where the setpoint reflects the ratio of the actuator’s and the annihilator’s synthesis rates.

Meanwhile, our analytical expressions for \bar{D} and $\bar{\lambda}$ should be treated as mere estimates. Indeed, in our manuscript’s lines 380–387 we acknowledge that even within our modeling framework they are only valid if the integral controller’s leakiness is neglected. Nonetheless, they can still provide

a qualitative insight of different circuit parameters' effect on the setpoints maintained by the controller.

1.1.6 Major Comment 6

To demonstrate the correct and robust functioning of the controller, the authors should perform an extensive set of numerical simulations. For instance, parameters of the coarse-grained model should be varied significantly (especially considering that the fit is not sensitive to most of these parameter values, see Supplementary Figure S2), and numerous disturbances should be considered, probing the effects of changes in the parameter values of the model as well as metabolic burden due to the expression of heterologous proteins. The scenarios should include not only steady-state but also time-varying burden, as well as stochastic analysis to demonstrate that noise does not compromise closed-loop performance.

Response: We would like to thank the Reviewer for their suggestions. Accordingly, in Supplementary Figure S6a–c we now display simulation results for our controller in presence of parameter uncertainty, sampling the parameter values from the probability distribution that we had fitted to data. We have also considered the case of the disturbing burdensome gene's expression oscillating over time (new Supplementary Figure S5), as well as simulated the circuit's stochastic performance as shown in Supplementary Figure S6d–f. These extensive simulations confirm the efficiency of our controller's performance for a wide range of conditions and model parameter values, which we now highlight in lines 372–379 in the main text's Results section.

1.1.7 Major Comment 7

Regarding the parameters of both the model and the controller, their values and units raise multiple issues.

- *Parameters of the model fit exhibit considerable “sloppiness” (Supplementary Figure S1), which may suggest that the model could be further simplified to eliminate this. The authors should explore the manifold that the potential combinations are constrained on to reveal their interdependencies, and to further reduce the model order and/or the number of free parameters.*

Response: We would like to thank the Reviewer for pointing out the possibility of further constraining the model's parameters. To investigate whether the fitted parameter space can be

constrained by defining one parameter as a function of another, similarly to what we did for α_a and α_r , we considered different pairwise combinations of parameter values in the newly added Supplementary Figure S2e. However, no clear relation emerged, pointing to the need to fit all four parameters in our Markov Chain Monte Carlo simulation.

Meanwhile, the model’s variables were carefully chosen to facilitate resource-aware gene circuit design, as we outline in the main text’s Discussion (lines 428–436 and 445–451). Therefore, while reducing the model’s order would indeed grant us greater simplicity, it risks abstracting the processes that are vital for analyzing the performance of biomolecular controllers and the effect of burden on the host cell’s state.

- *The typical range of all parameters should be sourced from experimental papers to ensure that the selected values represent realistic choices. Similarly, considering the simulated trajectories of the model (with and without the controller), the range of all species that appear in the model should also be evaluated against their typical ranges (this could also aid the selection of fitted parameters to ensure realistic parameterization).*

Response: Regarding the parameters being fitted, we have expanded the main text’s Methods section (lines 636–639) with an explanation of how we used estimates from published literature to choose the priors and admissible ranges for the values of the parameters being fitted. Supplementary Table S3 now includes a new column with citations of the publications from which the estimate was taken.

Comparing the simulated dynamic trajectories to experimentally established typical ranges is problematic. For the variables describing the host cell itself, few experimentally measured trajectories over time exist, with most studies focusing instead on steady-state observations – an exception to this are nutrient shift dynamics, a case of which we now include and compare to experimental data in the newly added Supplementary Note S1.6. Nonetheless, our work includes multiple evaluations of steady-state model predictions for different cellular variables (see Figure 3a–d and Figure 4e in the main text).

When simulating our controller, given that this is a novel design that has not yet been implemented *in vivo*, it is difficult to identify the “typical ranges” within which its dynamic trajectories would be contained. However, as a general rule, we have made sure that the overall mass fraction of all circuit proteins in the cell (i.e., heterologous protein mass fraction) never reaches the threshold of 0.25. As it is demonstrated in Figure 5, this value has been experimentally shown to

be feasible in *E. coli*, whereas for heterologous protein mass fractions below this value our model's predictions for the relationship between burden and growth are highly consistent with experimental data. We have also ensured that the parameters used to simulate our proposed controller and all other circuits fall within realistic experimentally established ranges. These ranges' boundaries and an explanation of how they were calculated can be found in the newly added Supplementary Note S4.1.

- *The units of several parameters appear puzzling. For instance, the variable T is supposed to be dimensionless, whereas in Supplementary Table 2 it is indicated to be measured in nM . In the same table, the unit of ν is $1/h$, whereas it should be nM/h . This issue persists throughout the Supplementary Information. The authors should carefully check all variables and parameters to ensure not only that their values are realistic, but also that their units are correct.*

Response: Thank you very much for pointing out these typos. We have checked the units throughout the Supplementary Information and, where needed, corrected and highlighted them in green.

1.2 Minor Comments

1.2.1 Minor Comment 1

While the Abstract is well-written, it should be trimmed in length.

Response: Thank you for your feedback on the abstract. We have now trimmed the abstract to make it meet the journal's editorial guidelines.

1.2.2 Minor Comment 2

The "Contents" on page 3 should be removed

Response: The "Contents" page has been removed from the manuscript.

1.2.3 Minor Comment 3

There is a typo in (5): in the argument of ν the variable s should be σ .

Response: We would like to thank the Reviewer for spotting this typo. We have corrected it, highlighting the amended piece in green.

1.2.4 Minor Comment 4

There are unresolved references in the Supplementary Information (e.g., on page 15).

Response: We have now checked the manuscript for any unresolved references and corrected them where necessary.

2 Reviewer 2

2.1 Major Comments

2.1.1 Major Comment 1

The authors model ppGpp regulation phenomenological through their parameter T . The authors' model only considers regulation of ribosomal production and tRNA production to be a function of T . ppGpp generally promotes transcription and can have significant transcription of amino acids as part of the "stringent response" (e.g. see <https://doi.org/10.1016/j.tim.2005.03.008>, <https://doi.org/10.1128/9781555816841.ch14>, and specific papers such as <https://doi.org/10.1073/pnas.0501170102>). While the authors' model (as is) fits experimental data in Figure 2, the authors should explain why they have not made F_a a function of T ? What impact do the authors think this would have on their model's findings?

Response: In Major Comments 1 and 2 the Reviewer raises points on the identity and regulation of the metabolic gene class. Given the interconnectedness of these concepts, please refer to our response to Major Comment 2 for a combined discussion on the first two Major Comments.

2.1.2 Major Comment 2

In equation 13, the authors show that "metabolic proteins aminoacylate the uncharged tRNAs". Are the authors referring to amino acid producing enzymes or the aminoacyl-tRNA synthetases? In Weisse et al. 2015, the authors assume these accessory proteins form part of their ribosomal species as do Liao et al. in their 2017 paper. My understanding is that some aminoacyl-tRNA synthetases are co-transcribed from their amino acid synthesis operon – making them "enzymes" rather than "ribosomes". The authors should explain their rationale for this equation and highlight any differences from previous works to aid the reader in setting this work in context.

Response: Our model implements bacterial resource allocation via ppGpp signaling according to the Flux-Parity Regulation Theory (Chure and Cremer 2023). This framework is formulated for the cases when nutrient concentrations in the environment are saturating (i.e., nutrients are in excess), so the tRNA aminoacylation flux is predominantly determined by the rate of tRNA aminoacylation by aminoacyl-tRNA synthetases (aaRSs) as defined in Equation (13). This makes a natural case for aaRSs being classified as metabolic genes, which are defined as the gene class enabling the tRNA aminoacylation flux.

Several previous publications, such as Scott et al. 2010, Weisse et al. 2015 and Liao et al. 2017, conversely, treat aaRSs as a part of the “extended ribosome” class of proteins, whose expression is co-regulated with ribosomal genes. However, experiments show that aaRS gene regulation is inconsistent with this assumption: unlike ribosomes, aaRS gene transcription exhibits no or weak dependence on ppGpp concentration (see Giegé and Springer 2016 and Blumenthal et al. 1976). Relegating aaRSs to the highly ppGpp-dependent ribosomal, rather than metabolic, gene class would be in contradiction with these findings.

Therefore, aaRSs, which are the key component of the metabolic gene class in our model, are assumed to not be regulated by ppGpp. Moreover, the metabolic gene class’s behavior as a whole is consistent with its transcription not being regulated by ppGpp (see Scott et al. 2010). This argument in favor of our assumption that F_a is not dependent on ppGpp concentration has been added to our manuscript in lines 495–499.

While the metabolic gene class as a whole can be treated as a lumped constitutive gene, it is still possible that some individual members of the ostensibly constitutive metabolic gene class are, for instance, upregulated in specific conditions. However, such changes would be negligible on the level of the entire gene class and/or compensated by other metabolic genes’ downregulation. In particular cases when the expression dynamics of some individual (e.g., nutrient-specific) metabolic genes becomes defining for the cell’s state, our model may have to be extended in order to provide a more nuanced view of bacterial physiology. In Supplementary Note S1.6, newly added in response to Reviewer 1’s Major Comment 1, we demonstrate how this can be done rather straightforwardly for the example of nutrient upshifts. A more fine-grained view of different metabolic genes’ regulation is beyond the scope of the current study, but may be achieved by treating the particular metabolic genes of interest separately from the overall coarse-grained a -class. ODEs describing the expression of such genes can be defined along the lines of Equations (1)-(2) for synthetic genes

introduced to the host, with the gene transcription regulation function F made to depend on the cellular variables and the culture conditions.

2.1.3 Major Comment 3

Please can the authors explain the biological rational of their parameter sigma (which governs efficiency of tRNA charging) and explain why they vary it between 0.08 and 0.5 in Figure 2.

Our model treats the aminoacylation of tRNAs using the medium’s nutrients as a single step with Michaelis-Menten kinetics. As mentioned in our response to reviewer 2’s Major Comments 1 and 2, this concerns the case of nutrients being present in excess, so their concentration does not affect this reaction’s rate. The dependence of cell growth rate on the culture medium therefore arises from the “quality” of its nutrients, i.e. the yield of aminoacyl-tRNAs obtained by metabolizing a nutrient molecule. Lower quality nutrients therefore typically yield fewer aminoacyl-tRNAs for the same concentrations of metabolic protein and uncharged tRNA. This scales the maximum rate of tRNA aminoacylation ν_{max} , giving rise to the interval of values $0 \leq \sigma \leq 1$, so nutrients with the highest possible yield give rise to the “true maximum” aminoacyl-tRNA synthesis rates $\nu_{max} \cdot \sigma = \nu_{max} \cdot 1 = \nu_{max}$.

Our coarse-grained model abstracts the particular pathways of amino acid synthesis. There is also no direct one-to-one link between the amino acid yield and the carbon content of a nutrient molecule, as some of the nutrient should contribute to ATP synthesis in order to cover the energy costs of aminoacyl-tRNA synthesis. Therefore, defining σ values for particular media is non-trivial. For media with known steady-state growth rates, one can simulate the model for different values of σ between 0 and 1 and pick the value yielding the closest steady-state growth rate. This, for example, is what we do in the newly added Supplementary Note S1.6.

This approach is not suitable when the model cannot yet be simulated due to some of its parameters being unknown. This is the case for our parameter fitting (outcome displayed in Figure 3, formerly Figure 2) to the data from Scott et al. 2010, which measured the cells’ steady-state growth rate and composition in culture media with different nutrient sources. The same data were also used to fit the model in Weisse et al. 2015, where the authors similarly faced the problem of quantifying the nutritiousness of qualitatively different media, which they denoted as the “nutrient efficiency factor” n_s . Their n_s factor is functionally similar to sigma in our model, as it scales the rate of ATP production, whose concentration in Weisse’s model determined the

translation elongation rate. In our case, the translation elongation rate is a function of aminoacyl-tRNA concentration, whose rate of production is scaled by σ . For the six different media in the experiment, Weisse et al. chose their n_s values as six log-spaced points between 0.08 and 0.5 as a convention. For the sake of consistency, we adopted the same values for our σ .

In the manuscript, we have now expanded the explanation of what σ represents (lines 547–553) and why during parameter fitting we made its values match those used in Weisse et al. 2015 (lines 613–616).

2.1.4 Major Comment 4

Regarding the demonstration of their models in demonstrating the experimentally verified Winner-Takes-All phenomenon, please can the authors highlight which model has been used in the main text.

Response: Following the Reviewer’s suggestion, we have now expanded the main text (lines 205–209), specifying that the expression of two self-activating genes in the “winner-takes-all” case, as well as all other instances of synthetic gene expression by the host, were simulated using the Supplementary Information’s Model VI with the generic synthetic gene expression ODEs substituted by the specific ODEs describing the circuitry in question. Captions to Figures 4, 5 and 7–9 now also refer to the Supplementary Information’s sections that provide the specific ODEs used to produce the displayed results.

2.1.5 Major Comment 5

The power of the authors’ analytical model is unclear to me. They define the protein production and growth rate in Eq. 23 and 24 which make reference to non-burdened values “NB”. This enables an analysis <relative> to the unburdened cell. The authors demonstrate the use of this two-state analytical model in understanding cell growth and burden as well as identifying optimal production rates for a single heterologous protein. Can the authors demonstrate or discuss how their findings could be used to understand a more complex process? Robust analytical methods would be a powerful addition to the field but here it appears the system is only tractable as the problem is so simple.

Response: The analytical methods that we propose indeed rely on knowing the steady state of all variables for a non-burdened cell (not expressing any synthetic genes) in given culture conditions. However, these values can be easily retrieved by simulating the host cell model without

any synthetic gene expression for a given value of the culture medium’s nutrient quality σ . To make this clear to the reader, amendments have been introduced to the main text’s lines 241–243.

The proposed analytical methods have also served to obtain predictions about the behavior of our integral controller for countering cell-wide fluctuations in ribosome availability. These relations are derived in the Supplementary Note S4.4.2, while the values calculated according to them are plotted as dotted lines and compared to numerical model predictions in the main text’s Figure 7 (formerly Figure 5). We would like to thank the Reviewer for highlighting the need to make this clearer in our manuscript, which we have done in lines 350–351.

Using the cell modeling framework and the associated analytical methods described in this article to predict the burden-dependence of other synthetic gene networks’ performance is also one of our current directions of further research. However, given the extent of the work carried out and the need to introduce additional context to explain the motivations behind the novel resource-aware controller designs in development, we believe that this lays outside of the scope of the present article.

2.1.6 Major Comment 6

In Eq 27 and the preceding sentence, please can the authors explain why the mRNA/ribosome dissociation constant is k_{poi}^{NB} (i.e. related to the unburdened system?)

Response: Analytical derivations in our model are enabled by the simplifying assumption that the translation elongation rate ϵ and the ribosomal gene transcription regulation function F_r are independent of synthetic gene expression burden. As we now elaborate in the main text (lines 235–237), this assumption is backed up both by numerical results and mathematical derivations, provided in the Supplementary Information’s Figure S3 and the newly added Supplementary Note S3.3. When obtaining analytical predictions, this allows us to treat these two variables as constant parameters, whose values can be retrieved by simulating the cell model without any burden for a given culture medium nutrient quality σ . These constant values are denoted as $\bar{\epsilon}^{NB}$ and \bar{F}_r^{NB} .

The mRNA-ribosome dissociation constant k_i for any gene, including $i = poi$ in Equation (27), is defined as a function of the translation elongation rate ϵ . Hence, our simplifying assumption that ϵ equals an unchanging constant $\bar{\epsilon}^{NB}$ also turns k_i into a constant \bar{k}_i^{NB} , which we explicitly show in the Supplementary Information’s Equation (S94). Given that the main text’s Equation (27) is

the result of analytical derivations for the gene of interest poi , the assumption that $k_{poi} = \bar{k}_{poi}^{NB}$ also applies to the case in question.

To elucidate this in the main text, we have added a short clarification for why the values of mRNA-ribosome dissociation constants are assumed to be equal to their “unburdened” values (lines 237–241), as well as restated that our derivations for maximizing heterologous protein production are enabled by the simplifying assumptions that we previously made (lines 350–351).

2.1.7 Major Comment 7

The authors’ PI controller is interesting. Can the authors show the open loop dynamics in Figure 5b–f to aid readers understanding the controller mechanism? The controller acts to maintain growth rate (mitigates gene expression burden) without interfacing with ribosome biogenesis or needing new regulatory linkages. It looks to me that this system could be described as a controlled ribosomal sponge with p_{amp} sequestering ribosomes and then being tuned down by the PI controller through the act and sens modules. While this maintains the cell’s burden (and so growth rate), this seems like a burdensome controller – i.e. the controller is required to sequester significant resources (which are then released for utilization by the circuit genes dist). To increase the utility of their results (and facilitate future biological implementation) can the authors demonstrate the performance of their controller at various growth rates?

Response: We are most grateful for the Reviewer’s suggestions. Controlled sequestration and release of ribosomes is an illuminating way of describing our circuit, which we have now incorporated into the caption of Figure 6. The Reviewer is also correct that the controller is burdensome. However, an overall reduced resource availability is an inherent consequence of using negative feedback to mitigate resource competition couplings, which has also been employed in other controllers with the same aims, such as Darlington et al. 2018 or Huang et al. 2021. It is likewise commonplace to maintain the desired cell growth rate by controlling the extent to which burden or toxicity slows down the rate of cell division relative to its value without the controller (e.g., see Dinh et al. 2020 or Gutierrez Mena et al. 2022). Consequently, there is a trade-off where the controller’s imposition of additional burden on the cell is necessary to enable the robustness of cell-wide resource availability to perturbations.

Following the Reviewer’s advice on improving the presentation of our simulation results, we have added open-loop simulations to the main text’s Figures 7–9, as well as Supplementary Figures S5–

S7, providing a visual intuition of the benefits of employing our controller. Figure 8 in the main text also displays the results of simulating our controller in culture media of different nutrient quality, which give rise to different steady-state growth rates. The practical utility of our circuit is further demonstrated in Figure 9, which considers an example application of ensuring the context-independence of an inducible genetic module.

2.1.8 Major Comment 8

The authors demonstrate the controller functions well between 0 and 2 nM, but based on their assumption that 1 molecule per cell is 1 nM, this suggests the controller only functions for low burden proteins/circuits from chromosomally expressed genes. However the authors state in Table S7 that c_{dist} and c_x are 100 nM which would demonstrate a wider utility (with these higher values implying the system works for parameters equivalent to medium-to-high copy plasmids). Can the authors review how they present the performance of their controller to demonstrate its function over a range of biologically feasible α and c values (or demonstrate the function over $c = c \cdot \alpha$)?

Response: The range of heterologous protein concentrations between 0 and 2 nM in the figure was a typographic error, which we have now fixed. The disturbing protein transcription rates were in fact varied from $c_{dist}\alpha_{dist} = 0$ nM/h up to $c_{dist}\alpha_{dist} = 7.5 \cdot 10^5$ nM/h – i.e., for $0 \leq c_{dist} \leq 1500$ nM, the upper bound of which is characteristic for extremely high-copy number plasmids. We would like to thank the Reviewer for bringing this typo to our attention.

2.1.9 Major Comment 9

Please can the authors give further details of their data fitting approach (e.g. cost function/loglikelihood functions) which are currently only available via the github repository.

Response: We have expanded the Main text’s Methods (lines 636–644) and Supplementary Note S2.2 (lines 267–273 and 289–301) to give more details on our choice of the likelihood function and assumed prior distributions for the parameters in our MCMC fitting, as well as the motivations behind our choices.

2.1.10 Major Comment 10

In recent years there has been a proliferation of host-circuit interaction frameworks, including other developed by the authors, and a number are available as preprints. Many are cited by the authors,

and I would urge the authors to reconsider their introduction to ensure none are missed. Most of these frameworks are based on similar concepts of ribosomal limitations as the authors. Please can the authors to explain more clearly the additional benefits of this new approach.

Response: We would like to thank the reviewer for attracting our attention to the need to position our model more clearly among the wide range of frameworks for modeling host-circuit interactions. In response, the Introduction and Discussion (lines 104–121 and 412–460, respectively) have been rewritten significantly to highlight the features of our model that we believe make it uniquely suitable for developing resource-aware synthetic biology designs, such as the ease of obtaining analytical predictions for synthetic circuit behavior (which reconciled several empirically observed relations) and the mechanistic consideration of all relevant steps of gene expression, combined with the recently developed ppGpp regulation theories originally formulated for the models with a higher level of abstraction.

While the range of resource-aware bacterial cell models is indeed very vast, hence the chance of us still missing some models, we have expanded the Introduction to include references to several modeling frameworks that were previously unmentioned, such as Erickson et al. 2017, Calabrese et al. 2021, Lin and Amir 2018, Roy et al. 2021, Korem Kohaim et al. 2018 and Molenaar et al. 2009.

2.2 Minor Comments

2.2.1 Minor Comment 1

In total the authors propose six coarse grained models of host circuit interactions. It would aid communication if the authors would make clear in their manuscript specifically which model has generated which results.

Response: The caption of Figure 3 (formerly Figure 2) in the main text has been extended to clarify that the results were obtained using the basic cell model ODEs in panels b–e and using the Supplementary Information’s Model IV in panel a to account for chloramphenicol action.

All instances of synthetic gene expression (Figures 4,5, and 7–9 in the updated manuscript), including the “winner-takes-all” scenario that Reviewer 2 inquires about in Major Comment 4, were modeled using the Supplementary Information’s Model VI with the generic synthetic gene expression ODEs substituted by the specific ODEs describing the circuitry in question. In the main text, we have now added a passage that specifies this (lines 205–209). Captions of the main

text’s Figures 4, 5, and 7–9 have also been completed with references to the circuit-specific ODE definitions given in the Supplementary Information.

2.2.2 Minor Comment 2

Figure 1 shows ppGpp explicitly and elsewhere the authors refer to “ppGpp regulation in our model”. However the authors’ inclusion of ppGpp is phenomenological (as opposed to mechanistic, as in e.g. Liao et al., 2017) – ideally the authors should make clear this is a phenomenological description given the wide readership of the journal.

Response: An explanation that ppGpp concentration follows a phenomenological relation, which allows us not to model its synthesis and degradation explicitly, has been added to the main text (lines 561–564).

2.2.3 Minor Comment 3

The authors should detail in their methods how numerical steady states were identified; were simulations “just run for a long time” or was the derivative evaluated and time span of the simulation increased until the derivative was small?

Response: Thank you for highlighting the need to include this information in our manuscript. We have now added the relevant explanation in the main text’s Methods (lines 583–591). Briefly, the model was assumed to have reached a steady state when its simulation reached the end of a pre-specified time interval. However, for the sake of computational efficiency, simulations were terminated prematurely if the sum of squared changes in all of the model’s variables was smaller than 10^{-6} over 12 hours.

2.2.4 Minor Comment 4

Page 9. Should “s” in Equation 5 should be sigma?

Response: Thank you for pointing this typo, which has now been amended.

2.2.5 Minor Comment 5

Page 13. Reference to 2d should be 2e?

Response: Thank you for spotting this typographical error. We have fixed it, highlighting the correction in green.

2.2.6 Minor Comment 6

When discussing “our model achieves near-optimal steady state growth rates”, I am not sure if the authors are claiming this as a new finding and they could set this work in context more clearly by making reference to other works (e.g. by Hidde de Jong and Terrance Hwa which show theoretically and experimentally how ppGpp control of bacterial resource allocation seems to be the solution to an optimal control problem – the authors already cite some of these works but not in this section).

Response: This was not claimed to be a new finding, but rather a confirmation of concordance with prior studies, including those mentioned by the reviewer. We have disambiguated the corresponding sentence (lines 182–186).

2.2.7 Minor Comment 7

The authors could highlight the importance of their work clearer by comparing their lumped factor zeta with the J metric of Santos-Navarro et al. (ref 24) and the S and Q metrics of McBride and Del Vecchio (ref 6).

Response: When introducing our lumped translational burden factor, we added a discussion of its similarities and differences with the J and Q metrics from (ref. 24) and (ref. 6) in the main text’s lines 253–267.

The “resource sensitivity coefficient” S from (ref. 6) quantifies the change in the steady-state value of a gene’s transcription regulation function F upon fluctuations in burden and does not have a direct analogue in our framework. We would like to thank the Reviewer for highlighting this, as the derivation of a similar metric using our model presents an interesting direction for further studies.

Reviewers' Comments:

Reviewer #1:

Remarks to the Author:

The authors have substantially revised the manuscript, both in terms of presentation and by incorporating additional examples and simulation data to further substantiate their results.

Regarding the novelty and impact, the authors now explicitly state how their model enables resource-aware gene circuit design by focusing on the design parameters of synthetic gene circuits and their impact on gene expression rates. The proposed model enables this by modeling transcription and translation as separate processes, and by allowing for modeling multiple common regulatory mechanisms. Additionally, instead of focusing on the overall biomass, the authors' model considers gene expression at the single-cell level, striking a delicate balance between accuracy and low model order. As a result, the proposed framework offers a tool for resource- and host-aware design of biomolecular circuits and controllers. The presented simulation data now includes transient scenarios as well (e.g., the cell's behavior during a nutrient upshift), and compares the numerical outcomes with experimental measurements both in and out of steady state, extending the scope of the original manuscript. While the authors mention that they did not want to single out ref [21] in their comparison, they may want to reconsider this and explicitly point out the differences in the manuscript (as they did in their Response to Referees) considering the similar nature of the models.

The presentation of the results has been significantly improved as well, for instance, by the inclusion of Figure 2 introducing the modeling framework in a graphical fashion, and by relocating some of the details to the Methods section. The main text is still fairly dense and notation-heavy, potentially negatively impacting the accessibility of the results by a broad audience (and this is even more pronounced in the Supplementary Material, where following the arguments and checking for technical correctness is especially challenging). As for the Methods, I would encourage the authors to split the introduction of the model into multiple parts focusing on each functional subsystem separately (as in ref [21]), and to mention all assumptions explicitly and together in one place.

The section originally focusing on the winner-take-all example is now expanded with the inclusion of another circuit to illustrate the deployment of the framework. In this second example changes in growth rate represent a key factor: reduced cell growth due to metabolic burden can result in emergent bistability in a system that otherwise exhibits monostability. These two examples together highlight that the model presented in the manuscript successfully captures the impact of metabolic burden from a qualitative perspective. A minor question: how does f_1 and f_2 affect the timescales (see Figure 4 caption)?

Considering the PI controller, the module based on an antithetic motif sequesters vast quantities of resources by acting as a sponge, which it can release to mitigate the negative impact of metabolic burden originating from other sources. In my opinion, this remains a major concern, though the authors correctly point out that other implementations also suffer from this limitation. Additionally, the results have been revised to regulate the protein concentration directly, instead of the variable D , making it more directly applicable. Finally, the authors present additional simulation data focusing on the performance of the proposed controller in the presence of parameter uncertainty, time-varying metabolic burden, and stochastic gene expression, confirming the correct functioning of the proposed controller module.

The authors have also addressed the questions about their modeling choices, for instance, the inclusion of degradation rates, the proportionality of synthesis rates with growth rate, the consumption of aminoacyl-tRNAs, various terms in the Hill functions, and the effective independence of the translation elongation rate and ribosomal gene transcription regulation from heterologous gene expression rates. Additionally, the authors included additional references detailing their choice of the priors and admissible ranges for the values of the parameters being fitted.

Reviewer #2:

Remarks to the Author:

The authors have now thoroughly reviewed the existing literature and set their work in the wider context of the existing field. Their numbered points (lines 114-121) make clear their aims in comparison to previous approaches.

The authors now describe their different models much more clearly and their inclusion of their data fitting cost function will enable replication of this work by others. It is now much clearer that T is the ppGpp inverse in Figure 2. The authors have responded to all of my queries but the new work leads to two further queries, below.

Regarding the authors analytical results, the discussion of these is now much clearer. The use of their methods to identify the set point of their controller is a useful demonstration of the approach. I understand the authors response to my Major Comment 5, i.e. that they feel that further complex investigation would be needed to extend these results beyond simple gene circuits and I look forward to reading their future work.

Comments on the new text

1. The addition of Tan et al's 2009 observations is good addition but I am not fully convinced by the authors analysis, as is. Please can the authors justify why toxicity might impacts ν (Eq. S117) whilst I can see a logic for this, I would have thought that they would capture the gross toxicity by multiplying λ by a p_7 dependent Hill function. I do not feel that the authors have not demonstrated via numerical experiments that their model recapitulates the observations of Tan et al. For example, what happens as the authors vary K_{tox} to tune the toxicity? As the toxicity decreases, I would have expected bistability to be lost as in the original paper. Given the authors have now carried out stochastic simulations of their controller – see comments below – they could consider stochastically simulating this model to show the emergence of the mixed population phenomenon observed experimentally although I acknowledge this would be time consuming and may not enhance the text much further.

2. The analysis of the controller has significantly improved. I thank the authors for showing the open loop simulation which highlights the performance of the controller. Figure 9 shows the controller functions well to reject translational and metabolic disturbances. Are the lines in Figure 9d labelled correctly? The raw values in Figure 9b show $c_{dist} = 500$, $\sigma = 0.5$ to be below $c_{dist} = 0$, $\sigma = 0.5$ while the normalised $p_x/p_{x,0}$ figure in Figure 9d shows $c_{dist} = 500$, $\sigma = 0.5$ to be above the $c_{dist} = 0$, $\sigma = 0.5$ line? If the lines are labelled correctly, I do not understand this analysis. The new analysis in Figure S6 is valuable but the authors give no details of how they carried out the stochastic simulations in panels Fig S6d, e, and f. Whilst I don't disagree with the authors comments related to the benefits of their controller when considering other burdensome controllers (e.g. Darlington et al, 2018 or Huang et al., 2021), it may help the readers understanding of this contribution to include the rebuttal discussion in the main text given recent works such as 10.1038/s41467-022-34647-1 which maintain growth rate.

A coarse-grained bacterial cell model for resource-aware analysis and design of synthetic gene circuits: Response to Peer Review Comments

December 19, 2023

Foreword

We are glad to have been able to address the reviewers' initial round of review responses, and would like to thank them for providing subsequent feedback on our updated manuscript. We are also most grateful to the editor for overseeing and facilitating the revision process. Here, we provide point-by-point responses to the second set of remarks made by the reviewers. In our manuscript and Supplementary Information, the amendments made in response to these comments are displayed in green.

1 Reviewer 1

The authors have substantially revised the manuscript, both in terms of presentation and by incorporating additional examples and simulation data to further substantiate their results.

Regarding the novelty and impact, the authors now explicitly state how their model enables resource-aware gene circuit design by focusing on the design parameters of synthetic gene circuits and their impact on gene expression rates. The proposed model enables this by modeling transcription and translation as separate processes, and by allowing for modeling multiple common regulatory mechanisms. Additionally, instead of focusing on the overall biomass, the authors' model considers gene expression at the single-cell level, striking a delicate balance between accuracy and low model

order. As a result, the proposed framework offers a tool for resource- and host-aware design of biomolecular circuits and controllers. The presented simulation data now includes transient scenarios as well (e.g., the cell's behavior during a nutrient upshift), and compares the numerical outcomes with experimental measurements both in and out of steady state, extending the scope of the original manuscript. While the authors mention that they did not want to single out ref [21] in their comparison, they may want to reconsider this and explicitly point out the differences in the manuscript (as they did in their Response to Referees) considering the similar nature of the models.

The presentation of the results has been significantly improved as well, for instance, by the inclusion of Figure 2 introducing the modeling framework in a graphical fashion, and by relocating some of the details to the Methods section. The main text is still fairly dense and notation-heavy, potentially negatively impacting the accessibility of the results by a broad audience (and this is even more pronounced in the Supplementary Material, where following the arguments and checking for technical correctness is especially challenging). As for the Methods, I would encourage the authors to split the introduction of the model into multiple parts focusing on each functional subsystem separately (as in ref [21]), and to mention all assumptions explicitly and together in one place.

The section originally focusing on the winner-take-all example is now expanded with the inclusion of another circuit to illustrate the deployment of the framework. In this second example changes in growth rate represent a key factor: reduced cell growth due to metabolic burden can result in emergent bistability in a system that otherwise exhibits monostability. These two examples together highlight that the model presented in the manuscript successfully captures the impact of metabolic burden from a qualitative perspective. A minor question: how does f_1 and f_2 affect the timescales (see Figure 4 caption)?

Considering the PI controller, the module based on an antithetic motif sequesters vast quantities of resources by acting as a sponge, which it can release to mitigate the negative impact of metabolic burden originating from other sources. In my opinion, this remains a major concern, though the authors correctly point out that other implementations also suffer from this limitation. Additionally, the results have been revised to regulate the protein concentration directly, instead of the variable D , making it more directly applicable. Finally, the authors present additional simulation data focusing on the performance of the proposed controller in the presence of parameter uncertainty, time-varying metabolic burden, and stochastic gene expression, confirming the correct functioning

of the proposed controller module.

The authors have also addressed the questions about their modeling choices, for instance, the inclusion of degradation rates, the proportionality of synthesis rates with growth rate, the consumption of aminoacyl-tRNAs, various terms in the Hill functions, and the effective independence of the translation elongation rate and ribosomal gene transcription regulation from heterologous gene expression rates. Additionally, the authors included additional references detailing their choice of the priors and admissible ranges for the values of the parameters being fitted.

Response: We are most grateful for the Reviewer’s positive feedback on our revisions and additions to the manuscript, and are glad that they now find our results, as well as their significance for the field of resource-aware synthetic biology, to be presented more clearly. We are likewise thankful for their further questions and suggestions how to further improve the manuscript’s clarity. Our response to these points is given below.

- Thank you for your suggestion to highlight the differences between our model and ref [21] in the main text’s Discussion. We now do this in the newly added lines 448–459, which make clear why we consider our model more suitable for the task of resource-aware synthetic gene circuit design compared to ref [21]. As suggested by the Reviewer, we hope that by clarifying how our work compares to the past literature, this will make our study more accessible to a broad audience.
- We thank the Reviewer for their valuable advice on how to further improve our model description. Following this advice, we have assembled a summary of all our modeling assumptions in the newly added Table 1 in the main text. Moreover, as the reviewer suggested, we now present our model part-by-part in the updated “Model definition” section of the main text’s Methods, which has been split into thematic sub-sections with italicized subheadings in the style of Nature Communications publications (e.g. as in Rottinghaus et al. 2022).
- We appreciate the reviewer’s request for an explanation of the connection between a bistable switch’s inducer level f_i and the timescale of its activation. Briefly, an increased concentration of the inducer in the medium significantly shortens the time taken to reach a switch’s high-expression equilibrium. When two bistable switches are present in the cell, it is therefore the switch with the higher corresponding inducer concentration that becomes activated earlier and prevents the other from following it due to the “winner-takes-all” effect. To reflect this,

we have added a clarification of this point to the caption of Figure 4 and the Supplementary Note S4.2.2’s lines 754–774. The newly added Supplementary Figure S4a illustrates the link between inducer levels and the switch activation timescales. Moreover, the key role that the timescales of activation play in the “winner-takes-all” behavior is now illustrated by Supplementary Figures S4b–d.

Additionally, to highlight how the “winner-takes-all” phenomenon qualitatively affects the system’s steady states, we have adjusted the simulation parameters in Supplementary Table S6 and altered the style of the plots in the main text’s Figure 4b and Supplementary Figure S4e. As opposed to plotting the equilibria for varied levels of f_1 with a single line, we now also show each fixed point’s position with a marker. This demonstrates more clearly that, unless the two switches are activated simultaneously, the “loser” switch protein’s concentration stays almost unchanged relative to its uninduced state, indicating that it is kept in its low-expression equilibrium by resource competition from the “winner switch”.

2 Reviewer 2

The authors have now thoroughly reviewed the existing literature and set their work in the wider context of the existing field. Their numbered points (lines 114-121) make clear their aims in comparison to previous approaches.

The authors now describe their different models much more clearly and their inclusion of their data fitting cost function will enable replication of this work by others. It is now much clearer that T is the $ppGpp$ inverse in Figure 2. The authors have responded to all of my queries but the new work leads to two further queries, below.

Regarding the authors analytical results, the discussion of these is now much clearer. The use of their methods to identify the set point of their controller is a useful demonstration of the approach. I understand the authors response to my Major Comment 5, i.e. that they feel that further complex investigation would be needed to extend these results beyond simple gene circuits and I look forward to reading their future work.

2.1 Comment on the new text 1

The addition of Tan et al’s 2009 observations is good addition but I am not fully convinced by the

authors analysis, as is. Please can the authors justify why toxicity might impacts ν (Eq. S117) whilst I can see a logic for this, I would have thought that they would capture the gross toxicity by multiplying λ by a p_{t7} dependent Hill function. I do not feel that the authors have not demonstrated via numerical experiments that their model recapitulates the observations of Tan et al. For example, what happens as the authors vary K_{tox} to tune the toxicity? As the toxicity decreases, I would have expected bistability to be lost as in the original paper. Given the authors have now carried out stochastic simulations of their controller – see comments below – they could consider stochastically simulating this model to show the emergence of the mixed population phenomenon observed experimentally although I acknowledge this would be time consuming and may not enhance the text much further.

Response:

- We thank the reviewer for their suggestion to elaborate on the motivation behind our choice to reflect the toxicity of T7 RNA polymerase (RNAP) by modifying the formula for the rate of the metabolic flux, i.e., the charging of tRNA molecules. We now do this in lines 821–828 of the Supplementary Information.

Multiplying λ by a p_{t7} -dependent Hill function, as it was done in the original publication by Tan et al., represents the naive approach of capturing the host-circuit interactions with a phenomenological relation between cell growth and the synthetic protein’s concentration. As we mention in the main text’s lines 419–424, this approach’s realism is limited, as it may abstract important aspects of cell physiology. In contrast, our cell model presents a more detailed picture of host-circuit interactions, allowing to capture the particular effects of burden on different cellular processes.

From this point of view, it is unlikely that the rate of cell growth is directly affected by T7 RNAP expression. The cell growth rate λ in our model is defined according to the constant proteome cellular trade-off (Weisse et al. 2015), which postulates that cell volume expands at a rate that maintains a constant protein mass density as confirmed by experimental studies like Basan et al. 2015. Thus, making λ explicitly depend on the T7 RNAP concentration would imply that its toxicity directly affects the regulation of cell division, for which we have found no evidence.

Conversely (and supporting our approach), a recent experimental study by Tan and Ng has shown that T7 RNAP-based gene expression systems reduce the host cell’s growth by

sequestering the cell’s ribosomes and tRNAs, as well as by interfering with amino acid metabolism. Since our resource-aware cell model explicitly considers ribosomal allocation and the consumption of aminoacyl-tRNAs during translation, it can by design capture the former two effects of T7 RNAP expression. Meanwhile, T7 RNAP’s negative effect on amino acid metabolism naturally calls for scaling the aminoacyl-tRNA production rate ν , which, as a coarse-grained variable, reflects the rate of amino acid production alongside the rates of nutrient import and the attachment of amino acid residues to uncharged tRNA molecules. While it is likely possible to propose a more fine-grained and therefore more accurate model for the mechanism of this metabolic interference, this would require introducing additional parameters and differential equations. Instead, in line with our goal of a model that balances realism and minimum model complexity, we have captured this effect using a simple Hill function.

- We would like to thank the Reviewer for their suggestion to investigate the effect of T7 RNAP’s toxicity, which we now define as γ_{tox} , an inverse of the previously used Hill constant K_{tox} . This change is meant to make our notation more consistent with that used by Tan et al., as well as render our analyses more intuitive to the reader, as a greater value of γ_{tox} means greater toxicity to the host cell. Moreover, lack of synthetic protein toxicity can now be modeled simply by setting $\gamma_{tox} = 0$ as opposed to requiring an infinitely high Hill constant K_{tox} .

Likewise to the observations of Tan et al., as we show in the newly added Supplementary Figure S5, little or no heterologous protein toxicity results in a wide range of initial conditions converging to a single equilibrium, hinting at a loss of bistability. Similar effects are observed when γ_{tox} is greatly increased. In order to obtain a fuller picture of the system’s stability, an expanded bifurcation analysis was performed according to the method from de Cesare et al. 2022. Described in Supplementary Note S4.3.2 and shown in Supplementary Figure S5d–g, this investigation allowed to estimate the bistable region’s boundaries in the parameter space, as well as confirmed monostability at low γ_{tox} values. Besides demonstrating our model’s consistence with prior findings, this study provides a blueprint for resource-aware bifurcation analysis of other synthetic gene circuits in the future.

In Supplementary Figure S5d–g we obtained the bifurcation diagram for protein levels p_{t7} , rather than mRNA concentrations m_{t7} . Therefore, please note that for the sake of consis-

tency we have re-selected initial conditions in the main text’s Figure 4d–e so as to start from different protein concentrations but the same mRNA levels, rather than vice versa as previously.

In summary, we explored how the circuit’s bistability is affected by T7 RNAP’s toxicity by simulating the system for different parameter values, as well as performing numerical bifurcation analysis. This analysis is now mentioned in the main text’s lines 226–228, whereas the main Figure 4d–e has been adjusted to complement it.

- Stochastic simulation of the non-cooperative self-activator in a cell population to reproduce the mixed population effects observed by Tan et al. is indeed an interesting use for our model. However, as the Reviewer has noted, this numerical experiment is likely to be computationally and time-intensive – especially because it requires simulating a population of multiple cells in parallel – while only tangentially concerning the main subject of our manuscript. We are therefore grateful for the Reviewer’s understanding of our choice not to include it in the present work.

2.2 Comment on the new text 2

The analysis of the controller has significantly improved. I thank the authors for showing the open loop simulation which highlights the performance of the controller. Figure 9 shows the controller functions well to reject translational and metabolic disturbances. Are the lines in Figure 9d labelled correctly? The raw values in Figure 9b show $c_{dist} = 500$, $\sigma = 0.5$ to be below $c_{dist} = 0$, $\sigma = 0.5$ while the normalised p_x/p_x^0 figure in Figure 9d shows $c_{dist} = 500$, $\sigma = 0.5$ to be above the $c_{dist} = 0$, $\sigma = 0.5$ line? If the lines are labelled correctly, I do not understand this analysis. The new analysis in Figure S6 is valuable but the authors give no details of how they carried out the stochastic simulations in panels Fig S6d, e, and f. Whilst I don’t disagree with the authors comments related to the benefits of their controller when considering other burdensome controllers (e.g. Darlington et al, 2018 or Huang et al., 2021), it may help the readers understanding of this contribution to include the rebuttal discussion in the main text given recent works such as 10.1038/s41467-022-34647-1 which maintain growth rate.

Response:

- We would like to thank the Reviewer for spotting the erroneous labeling in the main text’s

Figure 9, which had been caused by a pdf rendering error. We have now fixed the underlying issue, as well as introduced additional annotations to clarify the type of disturbance represented by each curve.

- Thank you very much for highlighting the need to describe our stochastic simulation approach in more detail, which we now do in the newly Supplementary Note S3.3 that explains our methods and includes a step-by-step summary of the simulation algorithm in Algorithm S1. Briefly, we use a hybrid model which treats the variables pertaining to heterologous genes (including those of our integral controller) as discrete and stochastic. Meanwhile, the host cell variables are considered continuous and deterministic, since they are coarse-grained and represent average dynamics of multiple individual variables, whose fluctuations can therefore be expected to average out (see Liao et al. 2017). Every $\Delta t = 10^{-6}$ hours, the deterministic update to the model’s variables is determined by ODE integration, while the changes caused by stochastic reactions are determined by sampling a Poisson random distribution. This hybrid tau-leaping simulation algorithm is similar to that in Hepp et al. 2015. A brief description of the above simulation approach has also been added to lines 590–596 of the main text’s Methods.

Reviewing the code during the preparation of Supplementary Note S3.3 has also allowed us to spot a mistake in our Matlab implementation of the hybrid simulation algorithm, which had previously made us slightly underestimate the extent of noise in stochastic expression of synthetic genes. The simulation results in Supplementary Figure S7d–f (formerly Supplementary Figure S6d–f) have therefore been recreated using the corrected script. Notably, our original observations remain true for the updated figure – namely, it can be seen that the controller, similarly to other circuits employing the antithetic integral feedback motif, achieves greater robustness to disturbances while increasing variance between the system’s stochastic trajectories (Briat et al. 2016).

- We are grateful for the Reviewer’s suggestions how to improve the description of our controller. A paragraph with the relevant discussion has therefore been added to the main text’s lines 407–415. As we did in our previous rebuttal letter, this new addition highlights the common trade-off of increasing burden to achieve greater robustness to perturbations. It also compares our controller to the one mentioned by the Reviewer and described in Barajas et

al. 2022. While the design in question is indeed less burdensome than ours, as a feedforward controller it relies on the disturbing gene and the controller gene being co-regulated. Meanwhile, our design has no such requirement and counters fluctuations in burden regardless of which particular gene causes them.

Reviewers' Comments:

Reviewer #1:

Remarks to the Author:

The authors have addressed my comments and suggestions about the comparison between their results and prior work, as well as the description and presentation of the model and the underlying assumptions. Regarding the latter, the authors should check the Methods section to ensure correct punctuation involving mathematical formulas and equations.

Reviewer #2:

Remarks to the Author:

I thank the authors for their time in clarifying aspects of their rebuttal.

I thank the authors for their expanded consideration of the work of Tan et al. The authors choice of "burden source" is well explained in the SM and their rebuttal on this point is interesting – I thank them for the discussion and the reference to Tan and Ng's work. I am pleased that the authors have extended their analysis by including the new Figure S5 which I think now puts their conclusions beyond doubt – it is now clear that their model captures the observations of Tan et al, i.e. as the toxicity factor γ_{tox} is increased the system shows bistability.

As I said in my previous comments, my only query regarding the new controller analysis related to the labelling of Figure 9. In light of the corrected labelling I have no further comments on this section. I thank the authors for their narrative comparison of the different controller approaches. I thank the authors for their new Supplementary Note S3.3 – I see no issues with the approach they have adopted.

A coarse-grained bacterial cell model for resource-aware analysis and design of synthetic gene circuits: Response to Peer Review Comments

Foreword

We are glad to have been able to address the reviewers' comments, and are most grateful to them for providing insightful feedback that has allowed us to improve our work. We would also like to thank the editor for facilitating the revision process.

1 Reviewer 1

The authors have addressed my comments and suggestions about the comparison between their results and prior work, as well as the description and presentation of the model and the underlying assumptions. Regarding the latter, the authors should check the Methods section to ensure correct punctuation involving mathematical formulas and equations.

Remarks on code availability: *The code provides a README file with sufficient instructions.*

Response: Thank you for your supportive feedback and your helpful comments to the earlier revisions. We have now reviewed the Methods section, along with all other sections of the main text and Supplementary Information, and corrected any punctuation errors and typos spotted.

2 Reviewer 2

I thank the authors for their time in clarifying aspects of their rebuttal.

I thank the authors for their expanded consideration of the work of Tan et al. The authors choice of “burden source” is well explained in the SM and their rebuttal on this point is interesting – I thank them for the discussion and the reference to Tan and Ng’s work. I am pleased that the authors have extended their analysis by including the new Figure S5 which I think now puts their conclusions beyond doubt – it is now clear that their model captures the observations of Tan et al, i.e. as the toxicity factor γ_{tox} is increased the system shows bistability.

As I said in my previous comments, my only query regarding the new controller analysis related to the labelling of Figure 9. In light of the corrected labelling I have no further comments on this section. I thank the authors for their narrative comparison of the different controller approaches. I thank the authors for their new Supplementary Note S3.3 – I see no issues with the approach they have adopted.

Remarks on code availability: *I have not run Sechkar’s code but I have looked through the contents of the repository and find it clearly laid out and see all the files I would expect to see for this paper.*

Response: We would like to thank the Reviewer for their positive response, as well as for their valuable suggestions in the previous stages of the review process. Before final submission, we proofread all of our code again to ensure legibility, eliminate typos, and make sure that all scripts can easily be run correctly by future users.